# On the VC dimension of deep group convolutional neural networks

**Anna Sepliarskaia**
Booking.com
Department of Applied Mathematics, University of Twente
Drienerlolaan 5
7522 NB Enschede, Netherlands
a.sepliarskaia@utwente.nl

**Sophie Langer**
Faculty of Mathematics
Ruhr University Bochum
Universitätsstraße 150
44801 Bochum, Germany
s.langer@rub.de

**Johannes Schmidt-Hieber**
Department of Applied Mathematics
University of Twente
Drienerlolaan 5
7522 NB Enschede, Netherlands
a.j.schmidt-hieber@utwente.nl

## Abstract

Equivariant neural networks outperform traditional deep neural networks on a number of tasks. The theoretical understanding of their generalization properties remains, however, limited. In this paper, we analyze the generalization capabilities of Group Convolutional Neural Networks (GCNNs) with ReLU activation function through the lens of Vapnik-Chervonenkis (VC) dimension theory. By deriving upper and lower bounds, we investigate how the network architecture affects the VC dimension.

## 1 Introduction

A central challenge in machine learning is selecting a model that generalizes well, meaning models that maintain high performance on both training and unseen data. In neural networks, a common strategy to enhance generalization is to explicitly incorporate task-specific symmetries into the network architecture, as is done in Group Convolutional Neural Networks (GCNNs). Therefore, understanding the theoretical aspects of generalization is particularly important for GCNNs—this is the problem we address in this paper.

GCNNs were first introduced by [10] to improve statistical efficiency and enhance geometric reasoning. Since then equivariant network structures have evolved to support equivariance on Euclidean groups [6, 5, 33], compact groups [18] and Riemannian manifolds [34]. More recent architectures have even been generalized beyond other types of symmetry groups [35, 11, 27]. GCNNs have demonstrated promise in a variety of domains, including fluid dynamics [32, 31], electrodynamics [35], medical image segmentation [6], partial differential equation (PDE) solvers [8], and video tracking [29]. A notable example of the success of equivariant neural networks is the AlphaFold algorithm, which achieves high accuracy in protein structure prediction through a novel roto-translation equivariant attention architecture of the neural network [15, 14].

While GCNNs have demonstrated strong empirical performance—often outperforming Deep (Feedforward) Neural Networks (DNNs)—rigorous theoretical guarantees remain limited. This gap is not due to a lack of interest; indeed, several works have attempted to address the theoretical understanding of GCNN's generalization. Most of these studies approach the problem by comparing the

39th Conference on Neural Information Processing Systems (NeurIPS 2025).

generalization behavior of equivariant neural networks to that of models without built-in symmetries [7, 24, 12, 26, 25, 28]. While these works offer valuable insights, they typically do not account for the specific GCNN architecture and are instead applicable to general equivariant or invariant classifiers. In contrast, our work provides generalization bounds for GCNNs that explicitly account for the number of layers, the number of weights, and the input dimension.

Recent works that study GCNN generalization for given architectures are, to the best of our knowledge, limited to two-layer networks [4, 21]. Our analysis extends this to GCNNs of arbitrary depth.

To study the generalization properties of GCNNs, we examine the VC (Vapnik–Chervonenkis) dimension, the pseudo-dimension, and the fat-shattering dimension—classical measures from statistical learning theory that quantify the capacity of a model class to fit arbitrary labelings. For example, a class that can realize all labelings of $n$ points has VC dimension at least $n$, while a class with only two different functions has VC dimension 1. Recent work has revived VC-based analysis for modern network architectures, including GCNNs [13, 21], convolutional neural networks (CNNs) [17], and DNNs [3].

## 2 Results

The main result establishes upper and lower bounds on the VC dimension of GCNNs with ReLU activation function. Let $\mathcal{H}_{W,L,r}$ and $\mathcal{F}_{W,L}$ denote the classes of GCNNs and DNNs, with ReLU activation function, number of weights $W$, depth $L$, and $r$ the group size in the GCNN. We prove that under the assumption $L < W^{0.99}$ there exist constants $c, C > 0$ such that

$$c\Big( \text{VC}(\mathcal{F}_{W,L}) + W \log_2(r) \Big) \leq \text{VC}(\mathcal{H}_{W,L,r}) \leq C\Big( \text{VC}(\mathcal{F}_{W,L}) + LW \log_2(r) \Big). \tag{1}$$

While we have not made an attempt to optimize the constants in the inequalities, Section C shows that if additionally $W > L^2$ and $W > 125$, one can take $c = 1/7500$ and $C = 500$.

This result demonstrates that, contrary to previous intuition, the class of GCNNs is richer than the class of DNNs with the same number of parameters. The difference in capacity can be expressed as a logarithmic function of the group size, with a larger group size leading to a more pronounced difference between the two classes. In the limiting case $r \to \infty$, the VC dimension of GCNNs diverges to infinity. Importantly, the analysis does not depend on the specific structure of the group and therefore applies to both continuous and compact Lie groups.

The analysis also naturally extends to the fat-shattering dimension, which measures how many points can be separated with a given margin, and thus additionally captures robustness of the classifier. For model classes that are closed under positive scaling, the fat-shattering and pseudo-dimensions coincide (see Theorem 11.14 in [1]). Since ReLU activations are positively homogeneous, that is, scaling the output of the final layer scales the entire network output, the GCNNs we consider are closed under positive scaling. Consequently, the generalization bounds derived in this paper for the pseudo-dimension also apply directly to the fat-shattering dimension.

Whereas the ReLU activation function is the running example, the inequalities hold under broader conditions on the activation function stated here (the constants $c, C$ do depend, however, on the activation function). As the proof of the upper bound follows the strategy in [3], the same assumptions appear. The lower bound holds if a DNN with this activation function can approximation the indicator function on an interval.

**Assumption 1** (Requirement for the upper bound). *The activation function is piecewise polynomial.*

**Assumption 2** (Range of activation functions for the lower bound). *For any $A < B$ and $\epsilon > 0$, there exists a DNN $\tilde{I}$ satisfying*

$$\tilde{I}(x) = \begin{cases} 1, & \text{if } x \in [A + \epsilon,\ B - \epsilon], \\ 0, & \text{if } x \notin [A - \epsilon,\ B + \epsilon]. \end{cases}$$

These assumptions imply that the same bounds hold for the Leaky ReLU $(x)_{+,\alpha} := \max(x, \alpha x)$, with $0 < \alpha < 1$ the Leaky ReLU parameter. Indeed, the Leaky ReLU is piecewise linear with two linear segments and therefore the upper bound applies. For the lower bounds, it is sufficient to check

that a DNN with Leaky ReLU activation can approximate the indicator function on $[A, B]$ using

$$\tilde{I}_{A,B}(x) := \frac{1}{(1-\alpha)\epsilon}\left[(x - (A - \epsilon))_{+,\alpha} - (x - A)_{+,\alpha} - \left((x - (B - \epsilon))_{+,\alpha} - (x - B)_{+,\alpha}\right)\right],$$

where $\epsilon > 0$, and $A < B - \epsilon$.

For general smooth activation functions, however, the current proof strategy does not extend directly. For instance, in the case of sigmoidal networks, a lower bound appears feasible because such networks can approximate indicator functions (see, e.g., Lemma 4 in [19]). In contrast, the upper bound does not follow readily: the region-counting argument, which applies to networks with piecewise polynomial activations like ReLU, fails for sigmoidal activations due to the absence of polyhedral decision boundaries.

Finally, to complete the comparison between the generalization bounds of GCNNs and DNNs, we also analyze the VC dimension of GCNNs with group size $r$ and DNNs when both architectures have the same number of neurons per layer. In this setting, GCNNs benefit from weight sharing across group elements, which reduces their number of parameters by a factor of $r^2$ compared to DNNs (see Section 4.2). We show that, under these conditions, the VC dimension of GCNNs is reduced proportionally to the group size $r$.

# 3 GCNNs

## 3.1 Group theoretic concepts

GCNNs capture symmetries in data via group theory. A *group* $G$ is a set equipped with an operation $\circ$ such that $h, g \in G$ implies $h \circ g \in G$; there exists an identity element $e \in G$ such that $e \circ g = g \circ e = g$ for all $g \in G$; there exists an inverse element $g^{-1} \in G$ such that $g^{-1} \circ g = g \circ g^{-1} = e$; and for any $g, h, i \in G$, we have $(g \circ h) \circ i = g \circ (h \circ i)$.

A **group action** describes how a group interacts with another set. More specifically, an *action* of $G$ on a set $\mathcal{X}$ is a map $\circ : G \times \mathcal{X} \to \mathcal{X}$ such that for all $g_1, g_2 \in G$ and for all $x \in \mathcal{X}$

$$(g_1 g_2) \circ x = g_1 \circ (g_2 \circ x).$$

On the functions $\{f : G \to \mathbb{R}\}$, the *left regular representation* of a group $G$ is the action

$$(g \circ f)(g') = f(g^{-1}g'), \quad \text{for any } g, g' \in G. \tag{2}$$

For a so-called kernel function $\mathcal{K} : G \to \mathbb{R}$ and a function $f : G \to \mathbb{R}$, the group convolution of $\mathcal{K}$ with $f$ at an element $g \in G$ is defined as

$$(\mathcal{K} \star f)(g) := \int_G \mathcal{K}(g^{-1}g') \cdot f(g') \, d\mu(g'), \tag{3}$$

with $\mu$ being the Haar measure [23, Theorem 8.1.2]. The Haar measure is a unique left-invariant measure that exists for all locally compact groups (see [30, Definition 1.18]). For the integral in (3) to be well-defined, we assume that both, the signal $f$ and the kernel $\mathcal{K}$, are measurable and bounded.

The group convolution computes a weighted average over the group elements. It is equivariant with respect to the left regular representation of $G$ defined in (2), that is, $g \circ (\mathcal{K} \star f) = (\mathcal{K} \star (g \circ f))$ for any group element $g$.

In the case where $G$ is a finite group, the group convolution becomes the sum

$$(\mathcal{K} \star f)(g) = \frac{1}{|G|} \sum_{g' \in G} \mathcal{K}(g^{-1}g') \cdot f(g'). \tag{4}$$

In practice, continuous groups such as rotations or translations are often discretized, and computations are performed on finite grids. This discretization process involves approximating the group $G$ by a finite subset

$$G^r = \{g_1, g_2, \ldots, g_r\}.$$

We refer to the cardinality $r$ as the resolution of the discretization.

For example, for the group formed by all continuous rotations, the discretization selects a finite number of rotation angles. For the translation group, discretization consists of a finite number of shifts.

After discretization, the group convolution operates on this finite set. Ignoring the reweighting it is then given by

$$(\mathcal{K} * f)(g) := \sum_{j=1}^{r} \mathcal{K}(g^{-1}g_j) \cdot f(g_j). \tag{5}$$

This operation is called *G-correlation* [9]. Note that $(\mathcal{K} * g) = r \cdot (\mathcal{K} \star g)$. If $\mathcal{K} = \mathbf{1}(\cdot = e)$ with $e$ the identity element of the group, then, on $G^r$, $K * f = f$. The $G$-correlation heavily depends on the discretization of the group and can differ significantly from the integral version (3). However, both definitions become approximately the same (up to rescaling) if the elements $g_j$ are drawn from the Haar measure. In the case of GCNNs, the discretization is determined by the structure of the data.

### 3.2 GCNN architecture

To parametrize the kernel function, let

$$\mathrm{K}_s : G \to \mathbb{R}, \quad s = 1, \dots, k \tag{6}$$

be a set of basis functions. The kernel functions $\mathcal{K}_\mathbf{w}$ are then expressed as linear combinations of these basis functions, that is,

$$\mathcal{K}_\mathbf{w} = \sum_{s=1}^{k} w_s \mathrm{K}_s, \tag{7}$$

where $\mathbf{w} = (w_1, \dots, w_k)$ is the vector of trainable parameters.

Given an activation function $\sigma : \mathbb{R} \to \mathbb{R}$, a group convolutional unit (GCNN unit) or G-convolution takes an input function on the discretized group $f = (f_1, \dots, f_{m_0})^T : G^r \to \mathbb{R}^{m_0}$ as input and outputs another function on the discretized group $h : G^r \to \mathbb{R}$. The output $h$ is

$$h = \sigma\left( \sum_{i=1}^{m_0} \mathcal{K}_{\mathbf{w}_i} * f_i - b \right), \tag{8}$$

with the group convolution operation $*$ defined in (5). The weight vectors $\mathbf{w}_1, \dots, \mathbf{w}_m$ and the bias $b$ are parameters that are learned from the data. In line with the common terminology, we refer to the output as feature map.

A GCNN layer (also called $G$-convolutional layer or $G$-conv layer) computes several GCNN units in parallel, using the same input functions but applying different kernel functions (also known as filters), each with potentially different parameters. Specifically, a GCNN layer with $M$ units and input function $f = (f_1, \dots, f_{m_0})^T : G^r \to \mathbb{R}^{m_0}$, computes the following $M$ functions

$$h_j = \sigma\left( \sum_{i=1}^{m_0} \mathcal{K}_{\mathbf{w}_{ij}} * f_i - b_j \right), \quad j = 1, \dots, M, \tag{9}$$

where $\mathcal{K}_{\mathbf{w}_{ij}}$ connects the $i$-th input with the $j$-th output. The trainable parameters in the GCNN layer are the weight vectors $\mathbf{w}_{ij}$ and the biases $b_j$. As other network architectures, GCNNs are typically structured hierarchically, with several GCNN layers followed by fully connected layers. We assume that the input of the first GCNN layer are already functions on the discretized group.

Let $L$ be the number of GCNN layers and assume that the respective numbers of GCNN units in the layers $1, \dots, L$ is denoted by $m_1, \dots, m_L$. To derive a recursive formula for the GCNN, we denote the inputs of the GCNN by $h_{0,1}, \dots, h_{0,m_0}$. Note that they are also assumed to be functions $G^r \to \mathbb{R}$. If for a given $\ell = 0, \dots, L-1$, $h_{\ell,1}, \dots, h_{\ell,m_\ell}$ are the outputs (also known as feature maps) of the $\ell$-th GCNN layer, then, the $m_{\ell+1}$ outputs of the $(\ell+1)$-st GCNN layer are given by

$$h_{\ell+1,j} = \sigma\left( \sum_{i=1}^{m_\ell} \mathcal{K}_{\mathbf{w}_{ij}^{(\ell)}} * h_{\ell,i} - b_j^{(\ell)} \right) \tag{10}$$

for $= 1, \ldots, m_{\ell+1}$. The final output of the GCNN is given by

$$\sum_{i=1}^{m_L} \sum_{g \in G^r} h_{L,i}(g), \tag{11}$$

which equals up to reweighting a global average pooling operation. As we take the sum over all group elements $g$, this operation makes the network invariant instead of equivariant to geometric transformation (see, e.g., [18, 16, 6]).

In this work, we consider the ReLU activation function $\sigma(x) = \max\{x, 0\}$. We denote the class of ReLU GCNNs with $L$ layers, $m_i$ units in layer $i = 0, \ldots, L$, $k$ dimensional weight vectors in (7), and $r$ the cardinality of the discretized group by

$$\mathcal{H}(k, m_0, \ldots, m_L, r). \tag{12}$$

The learnable parameters are the vectors $\mathbf{w}_{ij}^{(\ell)}$ and the biases $b_j^{(\ell)}$ for $j = 1, \ldots, m_\ell, \ell = 0, \ldots, L-1$. During the training phase, they are updated through gradient-based optimization techniques, such as stochastic gradient descent (SGD). The updates aim to minimize an objective function, typically a loss function measuring the difference between the network predictions and the true labels. While in practice, GCNN architectures also include feedforward layers, we only focus in this work on the GCNN layers.

### 3.3   CNNs as a special case of GCNNs

The convolutional layer in CNNs can be obtained as a specific case of the GCNN layer for the translation group $T$. An element $\mathbf{t}$ of the translation group corresponds to a vector $(t_1, t_2) \in \mathbb{R}^2$, and the group operation is defined as $\mathbf{t} \circ \mathbf{t}' = (t_1 + t_1', t_2 + t_2')$, meaning that one vector is shifted by the components of another. The inverse of a translation $\mathbf{t}$ is $\mathbf{t}^{-1} = (-t_1, -t_2)$, reversing the direction of the shift.

A square image can be interpreted as a function on the unit square $[0, 1] \times [0, 1]$. Setting the function values to zero outside the unit square, it can then be extended to a function on $\mathbb{R}^2$. Since $\mathbb{R}^2$ is isomorphic to the translation group, a square image can thus be viewed as a function on the translation group $T$.

To illustrate the discretization step, consider the MNIST dataset [20], which consists of grayscale images of handwritten digits ranging from 0 to 9. Each MNIST image is represented by grayscale values. This means that $m_0$ is equal to 1, as the image is characterized solely by pixel brightness. In contrast, for an RGB image, $m_0 = 3$, corresponding to the red, green, and blue components of each pixel.

Furthermore, since the values are only on a $28 \times 28$ pixel grid, the translation group $T$ is discretized by $T^{784} = \left\{ \left( \frac{i}{28}, \frac{j}{28} \right) \mid i, j \in [28] \right\}$. In turn, we can view an MNIST image as a function $f$ on $T^{784}$. The function value $f(\frac{i}{28}, \frac{j}{28})$ is the $(i, j)$-th pixel value.

In CNNs, the coefficients of the convolutional filters are typically represented by $s \times s$ weight matrices, with $s$ a prechosen integer. For simplicity, we consider $s = 3$ in the following. The convolutional filter computes $\sum_{\ell,k=-1}^{1} w_{\ell+1,k+1} f\left( \frac{i+\ell}{28}, \frac{j+k}{28} \right)$, $i, j = 1, \ldots, 28$. For the kernel $\mathcal{K}_{\mathbf{w}}(u_1, u_2) = \sum_{\ell,k=-1}^{1} w_{\ell+1,k+1} \mathbf{1}\left( (u_1, u_2) = \left( \frac{\ell}{28}, \frac{k}{28} \right) \right)$, and $\mathbf{s}_{i,j} := (i/28, j/28)$, this can also be rewritten in the form (5) via

$$(\mathcal{K}_{\mathbf{w}} * f)(\mathbf{s}_{i,j}) = \sum_{\mathbf{t} \in T^{784}} \mathcal{K}_{\mathbf{w}}(\mathbf{s}_{i,j}^{-1} \circ \mathbf{t}) \cdot f(\mathbf{t})$$

$$= \sum_{\mathbf{t} \in T^{784}} \sum_{\ell,k=-1}^{1} w_{\ell+1,k+1} \mathbf{1}\left( \left( t_1 - \frac{i}{28}, t_2 - \frac{j}{28} \right) = \left( \frac{\ell}{28}, \frac{k}{28} \right) \right) \cdot f(\mathbf{t})$$

$$= \sum_{\ell,k=-1}^{1} w_{\ell+1,k+1} f\left( \frac{i+\ell}{28}, \frac{j+k}{28} \right).$$

Thus, in this equivalence, the kernel is a linear combination of indicator functions.

### 3.4 Difference between GCNN and deep feedforward network architectures

GCNNs and deep feedforward neural networks (DNNs) differ in how their computational units are defined. In a DNN, each unit computes an affine transformations applied to the output of the previous layer, followed by an activation function $\sigma$. If $\mathbf{z}$ is the output of the previous layer, $\mathbf{w}$ is a weight vector, and $b$ a bias, then a unit in a DNN computes

$$\sigma\big(\mathbf{w}^\top \mathbf{z} - b\big). \tag{13}$$

The class of ReLU DNNs with $L$ layers (that is, $L-1$ hidden layers and one output layer), $m_i$ units (or neurons) in layer $i = 0, \ldots, L$, and a single unit in the output layer (i.e., $m_L = 1$) is denoted by

$$\mathcal{F}(m_0, \ldots, m_L). \tag{14}$$

While both the DNN class $\mathcal{F}(m_0, \ldots, m_L)$ and the GCNN class $\mathcal{H}(k, m_0, \ldots, m_L, r)$ share the same architectural structure and apply the ReLU activation function, they differ in their unit definition, meaning that $\mathcal{F}(m_0, \ldots, m_L)$ uses DNN units (13) and $\mathcal{H}(k, m_0, \ldots, m_L, r)$, employs GCNN units (8).

## 4 VC dimension of GCNNs

We now derive upper bounds for the VC dimension of the GCNN class $\mathcal{H}(k, m_0, \ldots, m_L, r)$. We begin by formally introducing the VC dimension.

**Definition 1** (Growth function, VC dimension, shattering). *Let $\mathcal{H}$ denote a class of functions from $\mathcal{F}$ to $\{-1, 1\}$ (often referred to as the hypotheses class). For any non-negative integer $m$, we define the growth function of $\mathcal{H}$ as the maximum number of distinct classifications of $m$ samples that can be achieved by classifiers from $\mathcal{H}$. Specifically, it is defined as:*

$$\Pi_{\mathcal{H}}(m) := \max_{f_1, \ldots, f_m \in \mathcal{F}} \left| \{(h(f_1), \ldots, h(f_m)) : h \in \mathcal{H}\} \right|.$$

*If $\mathcal{H}$ can generate all possible $2^m$ classifications for a set of $m$ inputs, we say that $\mathcal{H}$ shatters that set. Formally, if*

$$\left| \{(h(f_1), \ldots, h(f_m)) : h \in \mathcal{H}\} \right| = 2^m,$$

*we say $\mathcal{H}$ shatters the set $\{f_1, \ldots, f_m\}$.*

*The Vapnik-Chervonenkis dimension of $\mathcal{H}$, denoted as $\mathrm{VC}(\mathcal{H})$, is the size of the largest shattered set, specifically the largest $m$ such that $\Pi_{\mathcal{H}}(m) = 2^m$. If no such largest $m$ exists, we define $\mathrm{VC}(\mathcal{H}) = \infty$.*

The VC dimension cannot be directly applied to a class of real-valued functions, such as neural networks. To address this, we follow the approach in [3] and use the pseudodimension as a measure of complexity. The pseudodimension is a natural extension of the VC dimension and retains similar uniform convergence properties (see [22] and Theorem 19.2 in [1]).

**Definition 2** (pseudodimension). *For a class $\mathcal{H}$ of real-valued functions, we define its pseudodimension as $\mathrm{VC}(\mathcal{H}) := \mathrm{VC}(\mathrm{sign}(\mathcal{H}))$, where*

$$\mathrm{sign}(\mathcal{H}) := \{\mathrm{sign}(H - b) \mid H \in \mathcal{H}, \ b \in \mathbb{R}\},$$

*where $\mathrm{sign}(x)$ is $1$ for $x > 0$ and $-1$ otherwise. We write $\Pi_{\mathcal{H}}$ to denote a growth function of $\mathrm{sign}(\mathcal{H})$.*

For common parameterized function classes, the VC dimension is related to the number of parameters. For DNNs, it also depends on the network depth, as discussed in [3]. As proved in this paper, for GCNNs, the VC dimension further depends on the resolution. The following result provides an upper bound on the VC dimension of the GCNN class $\mathcal{H}(k, m_0, \ldots, m_L, r)$ formally defined in (12). Recall that $L$ is the number of layers, $m_i$ is the number of units in layer $i = 0, \ldots, L$, weight vectors are $k$ dimensional, and $r$ is the cardinality of the discretized group.

**Theorem 1** (Upper Bound). *The VC dimension of the GCNN class $\mathcal{H} = \mathcal{H}(k, m_0, \ldots, m_L, r)$ is upper bounded by*

$$UB(\mathcal{H}) := L + 1 + 4\left(\sum_{\ell=1}^{L} W_\ell\right) \log_2\left(8er\sum_{\ell=1}^{L} m_\ell\right), \tag{15}$$

with $W_\ell$ the number of parameters up to the $\ell$-th layer, that is,

$$W_\ell := \sum_{j=1}^{\ell} m_j(km_{j-1} + 1). \tag{16}$$

The proof strategy is borrowed from [3], and the detailed proof is provided in the supplement.

An alternative to bounding the VC dimension based on the number of layers and neurons per layer is to express the bound in terms of the total number of trainable parameters. The main advantage of this approach is that it applies to a wider range of architectures including sparsely connected GCNNs. In this context, [3] establishes a bound on the VC dimension for the class

$$\mathcal{F}_{W,L} := \{\mathcal{F} = \mathcal{F}(m_0, \ldots, m_\ell) \mid \ell \leq L, W_L(\mathcal{F}) \leq W\}, \tag{17}$$

consisting of DNNs with at most $L$ hidden layers and an overall number of weights $W$.

Similarly, we consider the GCNN class:

$$\mathcal{H}_{W,L,r} := \{\mathcal{H}(k, m_0, \ldots, m_\ell, r) \mid \ell \leq L, W_L \leq W\}, \tag{18}$$

consisting of all GCNN architectures with a total number of parameters bounded by $W$, depth at most $L$, and $r$ the cardinality of the discretized group. The following theorem provides a lower bound on the VC dimension for this class:

**Theorem 2** (Lower bound). *If $W, L > 3$, then there exists a universal constant $c$ such that*

$$\mathrm{VC}(\mathcal{H}_{W,L,r}) \geq c \max\{\mathrm{VC}(\mathcal{F}_{W,L}), W \log_2(r)\}.$$

The proof of this theorem is particularly challenging and requires novel techniques. Standard approaches used for DNNs cannot be directly applied, as the second term in the lower bound depends on the cardinality of the group—an aspect unique to GCNNs. Unlike DNNs, where the VC dimension is determined solely by the number of parameters and layers, GCNNs introduce an additional dependency on the input dimension. To overcome these challenges, we establish a new connection between GCNNs and DNNs and show how the structure of GCNNs allows indicator-based functions to shatter $\lfloor \log_2(r) \rfloor$ input functions. A proof sketch is provided in Section 6, with the full proof detailed in the supplement.

### 4.1 Analyzing the tightness of bounds for GCNNs

To compare the upper and lower bounds, we first establish a connection between the VC dimensions of DNNs and the upper bound for GCNNs, as $\mathrm{VC}(\mathcal{F}_{W,L})$ appears in the lower bound. For this, we rely on the VC dimension bound for DNNs with piecewise-polynomial activation functions, derived in Theorem 7 of [3]. Specifically, we focus on the class of DNNs with $L$ layers and $m_i$ units in layer $i$, as defined in (14). This network class corresponds to the case where $d = 1$ and $p = 1$ in Theorem 6 of [3]. By applying the inequality $\log_2(\log_2(x)) \leq \log_2(x)$, which holds for any $x \geq 2$, the VC dimension bound for this class simplifies to

$$UB(\mathcal{F}) := L + 2\left(\sum_{\ell=1}^{L} W_\ell(\mathcal{F})\right) \log_2\left(4e \sum_{\ell=1}^{L} \ell m_\ell\right),$$

where $\mathcal{F}$ is shorthand for $\mathcal{F}(m_0, \ldots, m_L)$. Here, $W_i(\mathcal{F})$ represents the number of parameters in $\mathcal{F}$ up to the $i$-th layer, given by

$$W_\ell(\mathcal{F}) = \sum_{j=1}^{\ell} m_j(m_{j-1} + 1). \tag{19}$$

Comparing (16) and (19), and assuming an equal number of computational units per layer in both GCNNs and DNNs, we observe that the number of parameters in GCNNs satisfies the inequality $W_i \leq kW_i(\mathcal{F})$, where $k$ is the dimension of the weight vector $\mathbf{w}$ in (7). Combining this with the bound in Theorem 1, we obtain

$$UB(\mathcal{H}) \leq 2UB(\mathcal{F}) + 4\left(\sum_{\ell=1}^{L} W_\ell(\mathcal{H})\right) \log_2(2r). \tag{20}$$

Combining this inequality with the Eq. (41) we arrive at the following result.

**Corollary 1.** *If $W$ is the total number of parameters and $L$ the depth of the network, with $L < W^{0.99}$, then there exists a universal constant $C$ such that* $\mathrm{VC}(\mathcal{H}_{W,L,r}) \leq C\left(\mathrm{VC}(\mathcal{F}_{W,L}) + LW\log_2(r)\right)$.

By combining Corollary 1 with Theorem 2, we derive the bounds presented in (1). In this setting, the difference between the upper and lower bounds on the VC dimension is at most a factor of $L$. Closing this gap is a challenging problem, but we conjecture that the lower bound could be improved. Our intuition stems from the fact that the upper bound is derived using a parameter-counting argument, closely following the approach in [3]. In that work, it was shown that when using a parameter-counting argument, the lower bound could be achieved, at least in the case of DNNs. This suggests that to fully close the gap, either the proof technique for the upper bound must be fundamentally changed, or a different example must be constructed to strengthen the lower bound.

In practical applications, GCNNs tend to have relatively small depth. For example, GCNNs use 7 layers for the rotated MNIST dataset and 14 layers for CIFAR-10 [9]. In this regime, the derived upper and lower bounds on the VC dimension are nearly identical, further supporting the practical tightness of the results.

### 4.2 Comparison of VC bounds for GCNNs and DNNs

Equivariant neural networks are expected to have lower sample complexity than DNNs without built-in symmetries, i.e those represented by $\mathcal{F}_r := \mathcal{F}(rm_0, \ldots, rm_L)$, where each layer $i$ has $rm_i$ units. As shown in the following lemma, compared to DNNs, GCNNs reduce the VC dimension and consequently the sample complexity proportionally to the group size $r$. This finding is consistent with previous work [7, 24].

**Lemma 1.** *For a neural network with $L$ layers and $W(\mathcal{F}_r)$ weights satisfying $L < W(\mathcal{F}_r)^{0.99}$, there exists $c > 0$ such that for the GCNN class $\mathcal{H} := \mathcal{H}(k, m_0, \ldots, m_L, r)$, the VC dimension satisfies*

$$VC(\mathcal{H}) \leq \frac{c}{r}VC(\mathcal{F}_r). \tag{21}$$

*Proof.* The dimension $k$ of the linear space generated by GCNN filters is bounded by $r$, as functions on $G^r$ lie in a subspace determined by the identifiers of $g \in G^r$. For DNNs where $L < W(\mathcal{F}_r)^{0.99}$, the VC dimension scales as $VC(\mathcal{F}_r) \asymp W(\mathcal{F}_r)L\log(W(\mathcal{F}_r))$, where $W(\mathcal{F}_r)$ represents the number of parameters in $\mathcal{F}_r$. Using Eq. (16), we obtain

$$VC(\mathcal{F}_r) \asymp W(\mathcal{F}_r)L\log(W(\mathcal{F}_r)) = \left(\sum_{\ell=1}^{L} rm_\ell(rm_{\ell-1}+1)\right)L\log\left(\sum_{\ell=1}^{L} rm_\ell(rm_{\ell-1}+1)\right)$$

$$\geq \left(\sum_{\ell=1}^{L} rm_\ell(km_{\ell-1}+1)\right)L\log\left(\sum_{\ell=1}^{L} rm_\ell(km_{\ell-1}+1)\right) = rW(\mathcal{H})L\log\left(rW(\mathcal{H})\right)$$

$$\geq O\left(rUB(\mathcal{H})\right).$$

Taking into account that $UB(\mathcal{H}) \geq VC(\mathcal{H})$, we obtain the claimed inequality (21). □

A different but relevant comparison considers GCNNs and DNNs with the same number of parameters. As stated in Theorem (2), in this case, the VC dimension of GCNNs is larger than that of DNNs. The difference lies in the scaling terms $W\log(r)$ and $WL\log(r)$ between $\mathcal{H}_{W,L,r}$ and $\mathcal{F}_{W,L}$.

## 5 Novelty compared to related work

In this section, we compare our contributions to the most relevant works on the generalization of neural networks. First, we consider [3], which analyzes the VC dimension of DNNs. While their results provide valuable insights, they do not account for the dependence on the resolution $r$ of the group acting on the input data. One of the contributions of our work is showing that the VC dimension of GCNNs depends logarithmically on $r$, a feature unique to GCNNs and absent in [3].

For the special case of $L = 1$ and $W = 2$, the results scale as $\log(r)$, aligning with the findings of [21]. However, we significantly extend these results to more practical settings, including deeper

architectures ($L > 1$) and larger parameter counts ($W > 2$), making the findings more applicable to real-world scenarios.

Finally, [26] studies the sample complexity of data augmentation and compares it to learning without augmentation. While their work is relevant to transformation-invariant learning, it does not analyze the VC dimension of neural networks. As a result, their findings address a different aspect of generalization and are complementary to ours.

## 6   Proof sketch of Theorem 2

To establish a lower bound for the VC dimension of the GCNN class, we recall that $\mathcal{H}_{W,L,r}$, as defined in (33), represents the class of GCNNs with resolution $r$, $k$ the dimension of the kernel space, at most $L$ layers, and no more than $W$ parameters. Additionally, let $G^r := \{g_1, g_2, \ldots, g_r\}$ be a discretized group containing the identity element $e$.

To prove Theorem 2, we establish two lower bounds on the VC dimension of the GCNN class: one in terms of the VC dimension of classical DNNs, and one in terms of the resolution $r$. We then combine these bounds to obtain the desired result. Specifically, we prove the existence of a universal constant $c > 0$ such that

$$\mathrm{VC}(\mathcal{H}_{W,L,r}) \geq c \cdot \mathrm{VC}(\mathcal{F}_{W,L}), \tag{22}$$

and

$$\mathrm{VC}(\mathcal{H}_{W,L,r}) \geq c \cdot W \lfloor \log_2 r \rfloor. \tag{23}$$

Here, $\mathcal{F}_{W,L}$ is the class of DNNs with at most $L$ hidden layers and at most $W$ weights, as defined in (34).

The proofs of both inequalities rely on the relationship between DNNs and GCNNs—specifically, that a GCNN whose convolutional filters are proportional to the indicator function of the identity element can be represented as a DNN. This result is important by itself and is formulated in Lemma 8 in the supplementary material. This representation further implies that in order to prove that GCNNs can shatter a set of $m$ functions, it is sufficient to construct DNNs with the same number of parameters that can shatter $m$ points. This observation is central to the proofs and guides the construction used in both bounds.

In both arguments, we utilize so-called indicator neural networks: DNNs with ReLU activations that approximate indicator functions over a specified interval $[a, b]$. The endpoints of these intervals serve as learnable parameters, allowing the network to adaptively define its support. Formally, this can be defined as follows:

$$\mathbf{1}_{(a,b,\epsilon)}(x) = \frac{1}{\epsilon}\Big(\big(x - (a - \epsilon)\big)_+ - \big(x - a\big)_+ + \big(x - b\big)_+ - \big(x - (b + \epsilon)\big)_+\Big), \tag{24}$$

with four neurons in the hidden layer and $(x)_+ := \max(x, 0)$ the ReLU activation function.

**Proof of bound** (22)   To prove (22), we begin by constructing a new class of DNNs by adding indicator networks to the original class $\mathcal{F}$, which shatters $\mathrm{VC}(\mathcal{F}_{W,L})$. These indicator components are designed to ensure that the resulting DNNs vanish outside a specified high-dimensional hypercube. This construction guarantees the existence of $m$ functions over $G^r$ for which the corresponding GCNNs produce the same outputs as the *original* class $\mathcal{F}$ on the points it shatters.

The combined networks use at most $5W$ weights and retain the same depth $L$. Finally, we use the fact that for any constant $t > 1$, the VC dimension satisfies $\mathrm{VC}(\mathcal{F}_{tW,L}) \geq c_t \cdot \mathrm{VC}(\mathcal{F}_{W,L})$ for some constant $c_t < 1$, completing the argument. Formally, this result is stated in the following lemma. The proof is deferred to the supplementary material.

**Lemma 2.** *For $L > 3$ let $\mathcal{H}_{5W,L+1,r}$ be the class of GCNNs defined as in (33) and $\mathcal{F}_{W,L}$ be the class of DNNs defined as in (34). Then $\mathrm{VC}(\mathcal{H}_{5W,L+1,r}) \geq \mathrm{VC}(\mathcal{F}_{W,L})$.*

**Corollary 2.** *In the setting of Lemma 2, if the number of layers $L > 3$ and $L \leq W^{0.99}$, then there exists a constant $c$ such that $\mathrm{VC}(\mathcal{H}_{W,L,r}) \geq c \cdot \mathrm{VC}(\mathcal{F}_{W,L})$.*

**Proof of bound** (23)   To prove the lower bound in (23), we consider a subclass of GCNNs represented by an *indicator* network. We show that functions from this class can shatter a set of input

functions $F_m \subset \{f : G^r \to \mathbb{R}\}$ with $m := \lfloor \log_2 r \rfloor$, by adjusting their parameters—specifically, the endpoints of a specified interval—to match the values of the input functions in $F_m$.

This GCNN class uses only 4 parameters and inherits the properties of 'indicator' networks; in particular, each function in the class outputs zero for any input function whose value lies outside a specified interval $[A, B]$. This ensures that the GCNN only shatters functions whose values fall within that interval.

Next, we consider the sum of $\tilde{W}$ 'indicator' networks, each designed to shatter functions with values in disjoint intervals. The resulting sum can shatter the union of these functions, meaning it can shatter $\tilde{W} \lfloor \log_2 r \rfloor$ functions in total. By choosing $W = 4\tilde{W}$, we conclude that $\mathrm{VC}(\mathcal{H}_{W,L,r}) \geq \frac{1}{4} W \lfloor \log_2 r \rfloor$, which completes the proof.

The following lemma provides a formal statement regarding the GCNN class represented by 'indicator' networks, while the corollary presents the result for the sum of $\tilde{W}$ 'indicator' networks. The proof can be found in the supplementary material.

**Lemma 3.** *Let $\mathcal{H}_{4,L,r}$ be the class of GCNNs defined as in* (33). *Then $VC(\mathcal{H}_{4,L,r}) \geq \lfloor \log_2 r \rfloor$. Moreover, for any two numbers $A < B$, there exists a finite subclass of GCNNs $\mathcal{H} \subset \mathcal{H}_{4,L,r}$ that shatters a set of $\log_2 r$ input functions $\{f_i : G^r \to [A, B] \mid i = 1, \ldots, \log_2 r\}$, and outputs zero for any input function $f : G^r \to \mathbb{R} \setminus [A, B]$.*

**Corollary 3.** *The VC dimension of the class $\mathcal{H}_{4W,L,r}$, consisting of GCNNs with $4W$ weights, $L$ layers, and resolution $r$ satisfies the inequality $\mathrm{VC}\left(\mathcal{H}_{4W,L,r}\right) \geq W \lfloor \log_2 r \rfloor$.*

By combining Corollaries 2 and 3, we obtain the lower bound

$$\mathrm{VC}(\mathcal{H}_{W,L,r}) \geq \max \left\{ c \, \mathrm{VC}(\mathcal{F}_{W,L}), \frac{1}{4} W \lfloor \log_2 r \rfloor \right\},$$

thereby proving Theorem 2.

# 7 Conclusion

In this work, we established nearly-tight VC dimension bounds for a class of GCNNs, showing that their complexity depends on the number of layers, the number of weights, and the resolution of the group acting on the input data. While GCNNs share similarities with DNNs, they exhibit an additional term, $W \lfloor \log_2 r \rfloor$, which grows logarithmically with the input resolution $r$. This aligns with prior findings [21] that, as $r$ approaches infinity, the VC dimension of GCNNs becomes unbounded.

The results highlight the sensitivity of GCNNs to group discretization, offering new insights into their model complexity and generalization behavior. While the upper and lower bounds are close in practical settings, a gap remains, and refining the lower bound remains an open challenge.

### Acknowledgements

J. S.-H. was supported in part by the NWO Vidi grant VI.Vidi.192.021. S.L. was supported in part by the NWO Veni grant VI.Veni.232.033

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

# A    Proof of Theorem 1

Recall that
$$\mathcal{H} = \mathcal{H}(k, m_0, \ldots, m_L, r), \tag{25}$$
where $r$ is the cardinality of the discretized group $G^r := \{g_1, g_2, \ldots, g_r\}$. The parameter $k$ determines the number of basis functions
$$\mathrm{K}_s : G^r \to \mathbb{R}, \quad s = 1, \ldots, k, \tag{26}$$
in the parametrization of the kernel function
$$\mathcal{K}_{\mathbf{w}} = \sum_{s=1}^{k} w_s \mathrm{K}_s.$$

The other parameters $m_0, \ldots, m_L$ define the network architecture, and $W_\ell$ represents the number of parameters in the GCNN up to layer $\ell$. The class $\mathcal{H}$ consists of all functions that can be represented by a neural network with this architecture.

We restate Theorem 1 for convenience:

**Theorem 3** (Theorem 1). *The VC dimension of the GCNN class $\mathcal{H} = \mathcal{H}(k, m_0, \ldots, m_L, r)$ with $r > 1$, is bounded from above by*
$$UB(\mathcal{H}) := L + 1 + 4 \left( \sum_{\ell=1}^{L} W_\ell \right) \log_2 \left( 8er \sum_{\ell=1}^{L} m_\ell \right). \tag{27}$$

For the proof, we consider an input consisting of $m$ functions from $G^r$ to $\mathbb{R}^{m_0}$, denoted by
$$F_m := \{f_1, \ldots, f_m\}. \tag{28}$$

To prove we use the following known result:

**Lemma 4.** *[Lemma 1, [2]] Let $p_1, \ldots, p_{\tilde{m}}$ be polynomials of degree at most $t$ depending on $n \leq \tilde{m}$ variables. Then*
$$\Pi := |\{(\mathrm{sign}(p_1(x)), \ldots, \mathrm{sign}(p_m(x))) : x \in \mathbb{R}^n\}| \leq 2 \left( \frac{2e\tilde{m}t}{n} \right)^n.$$

We denote $S(\ell)$ the number of regions in the parameter space $\mathbb{R}^{W_\ell}$, such that in each region, the GCNN units in the $\ell$-th layer (denoted by $\{h_{\ell,j}(g) \mid j \leq m_\ell, f \in F_m, g \in G^r\}$) behave like a fixed polynomial of degree at most $\ell$ in the $W_\ell$ network parameters that occur up to layer $\ell$.

**Lemma 5.** *Let $\mathcal{H}$ be the class of GCNNs defined in (25), with at most $W_\ell$ parameters up to layer $\ell \in \{1, \ldots, L\}$. If $F_m$ is the class of functions defined in (28), and $S(\ell)$ is as defined above, then for $\ell = 0, 1, \ldots, L-1$,*
$$S(\ell+1) \leq 2 \left( \frac{2em_{\ell+1}mr(\ell+1)}{W_{\ell+1}} \right)^{W_{\ell+1}} S(\ell). \tag{29}$$

*Moreover, the GCNN units $\{h_{\ell+1,j}(g) \mid j \leq m_{\ell+1}, f \in F_m, g \in G^r\}$ with $h_{\ell+1,j}$ defined for different functions $f \in F_m$, are piecewise polynomials of degree $\leq \ell + 1$ in the network parameters.*

*Proof.* As a first step of the proof, we show that any GCNN unit $h_{\ell,j}$ of any layer $\ell \in \{0, \ldots, L\}$ and $j \in \{1, \ldots, m_\ell\}$ is a piecewise polynomial of degree at most $\ell$. We proceed by induction on the layers $\ell$.

For the base case $\ell = 0$, the GCNN units $h_{0,j}$ for $j \leq m_0$ correspond to the input of the network. As the inputs are independent of the network parameters, $h_{0,j}$ are polynomials of degree $0$.

Assume the statement holds for all layers up to $\ell$. We now prove it for layer $\ell + 1$. The GCNN unit in layer $\ell + 1$ is defined by a convolution with the feature maps from the previous layer, that is,
$$h_{\ell+1,j} = \sigma \left( \sum_{i=1}^{m_\ell} \mathcal{K}_{\mathbf{w}_{ij}^{(\ell)}} * h_{\ell,i} - b_j^{(\ell)} \right),$$

where the convolutional filter is expanded in terms of the fixed basis functions $\mathrm{K}_s$ via $\mathcal{K}_\mathbf{w} = \sum_{s=1}^{k} w_s \mathrm{K}_s$. For fixed network parameters, $g \mapsto h_{\ell,j}(g)$ is a function of the group, with $h_{\ell,j}(g) \in \mathbb{R}$ for any $g \in G^r$. By the induction hypothesis, $h_{\ell,j}(g)$ are piecewise polynomials of degree at most $\ell$ with respect to the network parameters, with the polynomial pieces depending on the network input and the group element $g$.

Next, for any input and any group element $g$, the term $(\mathrm{K}_s * h_{\ell,j})(g)$ can be written as

$$(\mathrm{K}_s * h_{\ell,j})(g) = \sum_{g' \in G^r} \mathrm{K}_s\big(g^{-1} \circ g'\big) \cdot h_{\ell,j}(g').$$

Since $h_{\ell,j}(g')$ is a piecewise polynomial of degree at most $\ell$, it follows that $(\mathrm{K}_s * h_{\ell,j})(g)$ is also a piecewise polynomial of degree at most $\ell$. Thus, the convolution

$$(\mathcal{K}_\mathbf{w} * h_{\ell,j})(g) = \sum_{s=1}^{k} w_s (\mathrm{K}_s * h_{\ell,j})(g)$$

is a weighted sum of piecewise polynomials, which remains a piecewise polynomial. However, multiplying by the weights $w_s$ increases the degree of the polynomial, making it at most $\ell + 1$. Subtracting the bias term $b_j^{(\ell)}$ and applying the ReLU activation function may increase the number of pieces, but does not increase the degree of the polynomials. Therefore, for any input and any group element $g$, the GCNN unit $h_{\ell+1,j}(g)$ remains a piecewise polynomial with degree $\leq \ell + 1$. This completes the proof by induction.

Next, we show (29). Each GCNN unit in layer $\ell + 1$ is computed by

$$\sigma \left( \sum_{i=1}^{m_\ell} \mathcal{K}_{\mathbf{w}_{ij}^{(\ell)}} * h_{\ell,i} \right),$$

with $\sigma(x) = \max\{x, 0\}$ the ReLU activation function. As mentioned above, applying the ReLU function can increases the number of regions in the parameter space where the GCNN units behave as polynomials. This occurs, as the ReLU function either outputs the input itself (for positive values) or zero (otherwise). As a result its application decomposes each of the $S(\ell)$ regions of the parameter space in layer $\ell$ in multiple subregions. To bound this number of subregions, we need to count the number of possible sign pattern that can arise after applying the ReLU activation.

Fixing one of the $S(\ell)$ regions of layer $\ell$, by definition, all functions $h_{\ell,j}(g)$ are polynomials in the parameters of degree at most $\ell$. Each of the $m_{\ell+1}$ GCNN units in layer $\ell + 1$, is then also a polynomial of degree at most $\ell + 1$, leading to at most $m_{\ell+1} mr$ polynomials, where $m$ is the number of input functions defined in (28) and $r$ is the resolution. Applying Lemma 4 to the $\tilde{m} = m_{\ell+1} mr$ polynomials of degree $t = \ell + 1$ depending on $n = W_{\ell+1}$ parameters leads to at most

$$2 \left( \frac{2 e m_{\ell+1} mr(\ell + 1)}{W_{\ell+1}} \right)^{W_{\ell+1}}.$$

different sign patterns for each region of $S(\ell)$. This shows (29) and concludes the proof.

$\square$

**Lemma 6.** *Let $\mathcal{H}$ be the class of GCNNs defined in (25), with at most $W_\ell$ parameters up to layer $\ell \leq L$, and $m_\ell$ GCNN units in layer $\ell$. For any integer $m > 0$, the growth function of this class can be bounded by*

$$\Pi_\mathcal{H}(m) \leq 2^L \prod_{\ell=1}^{L} \left( \frac{2 e m r m_\ell \ell}{W_\ell} \right)^{W_\ell} 2 \left( \frac{2 e m L}{W_L + 1} \right)^{W_L + 1}.$$

*Proof.* Lemma 5 shows that after $L$ layers, there are at most

$$2^L \prod_{\ell=1}^{L} \left( \frac{2 e m r m_\ell \ell}{W_\ell} \right)^{W_\ell}$$

regions in the parameter space $\mathbb{R}^W$, on which the GCNN units in the last layer $\{h_{L,i}(g) \mid i \leq m_L, f \in F_m, g \in G^r\}$ behave like a fixed polynomial function of degree $\leq L$ in $W$ variables.

Recall that the final output of the neural network, is obtained by applying average pooling to the outputs of the GCNN units in the last layer. This implies that, for a fixed network architecture and input, the output of the neural network is a piecewise polynomial of degree at most $L$, depending on all $W_L$ network parameters. Since there are $m$ possible inputs $f_1, \ldots, f_m$, we get $m$ piecewise polynomials, each corresponding to one of these inputs. Bounding the growth function $\Pi_{\mathcal{H}}(m)$ now means we need to count the number of different sign patterns that arises for classifiers in $\mathrm{sign}(\mathcal{H})$. For that, we recall that by Definition 3.2 in the main article,

$$\mathrm{sign}(\mathcal{H}) := \{\mathrm{sign}(h_{\mathbf{w}} - b) | h_{\mathbf{w}} \in \mathcal{H}, \mathbf{w} \in R^{W_L}, b \in \mathbb{R}\}.$$

Applying Lemma 4 to $m$ polynomials of the form $h_{\mathbf{w}} - b$ of degree at most $L$ and $W_L + 1$ variables leads to no more than

$$2 \left( \frac{2emL}{W_L + 1} \right)^{W_L + 1} \tag{30}$$

distinct sign patterns that the classifiers in $\mathrm{sign}(\mathcal{H})$ can produce.

Thus, the growth function within each region, where the GCNN units in the last layer $\{h_{L,i}(g) \mid i \leq m_L, f \in F_m, g \in G^r\}$ behave like a fixed polynomial function in $W$ variables, is bounded by (30). As a result, we conclude that the overall growth function $\Pi_{\mathcal{H}}(m)$ is bounded by

$$S(L) \cdot 2 \left( \frac{2emL}{W_L + 1} \right)^{W_L + 1} = 2^L \prod_{\ell=1}^{L} \left( \frac{2emrm_\ell \ell}{W_\ell} \right)^{W_\ell} \cdot 2 \left( \frac{2emL}{W_L + 1} \right)^{W_L + 1}.$$

This completes the proof. $\qquad\qquad\square$

For the proof of the Theorem 3 we also use the following technical lemma

**Lemma 7.** *Suppose $2^{\tilde{m}} \leq 2^{\kappa} \left( \frac{\tilde{m} \cdot \tilde{r}}{\tilde{w}} \right)^{\tilde{w}}$ for some $\tilde{r} \geq 16$ and $\tilde{m} \geq \tilde{w} \geq \kappa \geq 0$. Then $\tilde{m} \leq \kappa + \tilde{w} \log_2(2\tilde{r} \log_2 \tilde{r})$.*

*Proof of Theorem 3.* Let $m := \mathrm{VC}(\mathcal{H})$. For convenience, define the sum

$$\tilde{W} := \sum_{i=1}^{L} W_i, \tag{31}$$

where $W_i$ denotes the number of parameters of a GCNN in $\mathcal{H}$ up to layer $i$.

We consider two complementary cases and prove the theorem for each of them separately.

**Case 1:** $m < \tilde{W} + W_L + 1$. In this case, we have $\tilde{W} + W_L + 1 < 3\tilde{W} < UB(\mathcal{H})$, where $UB(\mathcal{H})$ is defined in (27). For the latter inequality we use that $\log_2 \left( 8er \sum_{\ell=1}^{L} m_\ell \right) > 1$. Therefore, Theorem 3 holds.

**Case 2:** $m \geq \tilde{W} + W_L + 1$. Since $m$ represents the VC dimension of $\mathcal{H}$, it follows from the definition of the VC dimension (see Definition 3.1 in the main article) that $\Pi_{\mathcal{H}}(m) = 2^m$. Applying Lemma 6 gives us

$$\Pi(m) = 2^m \leq 2^{L+1} \prod_{\ell=1}^{L} \left( \frac{2emrm_\ell \ell}{W_\ell} \right)^{W_\ell} \left( \frac{2emL}{W_L + 1} \right)^{W_L + 1}. \tag{32}$$

Next, we apply the weighted arithmetic-geometric mean (AM-GM) inequality to the right side of (32), using weights $W_\ell/(\tilde{W} + W_L + 1)$ for $\ell = 1, 2, \ldots, L$, and $W_L/(\tilde{W} + W_L + 1)$, where $\tilde{W}$ is defined in (31). This yields

$$2^m \leq 2^{L+1} \left( \frac{2em(r \sum_{\ell=1}^{L} \ell m_\ell + L)}{\tilde{W} + W_L + 1} \right)^{\tilde{W} + W_L + 1}.$$

The last step of the proof involves applying Lemma 7 in [2] to this inequality, which provides an upper bound for $m$. Before doing so, we must verify that all conditions of the lemma are satisfied. In our case, $\tilde{m}$ corresponds to $m$, $\kappa$ to $L+1$, $\tilde{w}$ to $\tilde{W}+W_L+1$, and $\tilde{r}$ to $2e(r\sum_{\ell=1}^{L}\ell m_\ell + L)$. Since $r\sum_{\ell=1}^{L}\ell m_\ell + L > 2$, we have $\tilde{r} > 16$. Moreover, we are considering the case where $m \geq \tilde{W}+W_L+1$, and it is straightforward to verify that $\tilde{W}+W_L+1 \geq L+1 > 0$. Therefore all conditions of Lemma 7 in [2] are indeed satisfied and we obtain

$$m \leq (L+1) + 2\tilde{W}\log_2\left(4e\Big(r\sum_{\ell=1}^{L}\ell m_\ell + L\Big)\cdot\log_2\Big(2e\Big(r\sum_{\ell=1}^{L}\ell m_\ell + L\Big)\Big)\right).$$

To simplify this inequality, we use that for all $a \geq 1$, $\log_2(2a\log_2 a) = \log_2(2a) + \log_2(\log_2 a) \leq 2\log_2(2a)$. Substituting $a = 2e\left(r\sum_{\ell=1}^{L}\ell m_\ell + L\right)$, we note that $a \leq 4er\sum_{\ell=1}^{L}\ell m_\ell$ and obtain

$$m \leq (L+1) + 4\tilde{W}\log_2\Big(8er\sum_{\ell=1}^{L}\ell m_\ell\Big),$$

completing the proof of the theorem. $\qquad\square$

## B  Proof of Theorem 2

In this section, we provide the detailed proof of lower bound on the VC dimension, along with the proofs for Lemmas 2, 3, and their corresponding Corollaries 2 and 3.

The class

$$\mathcal{H}_{W,L,r} := \{\mathcal{H}(k, m_0, \ldots, m_\ell, r) \mid \ell \leq L,\, W_L \leq W\}, \tag{33}$$

includes all GCNN architectures with a total number of parameters bounded by $W$, a maximum depth of $L$, and $r$ representing the cardinality of the discretized group $G^r := \{g_1, g_2, \ldots, g_r\}$ containing the identity element $e$.

Next, we recall that $\mathcal{F}(m_0, \ldots, m_L)$ represents the class of fully connected feedforward ReLU networks with $L$ layers, where $m_i$ denotes the number of units in the $i$-th layer for $i = 1, \ldots, L$. The output of the last hidden layer of any neural network $\tilde{h}_\mathbf{w} \in \mathcal{F}(m_0, \ldots, m_L)$, with parameters $\mathbf{w}$, can be written as a vector of size $m_L$, that is, $(\tilde{h}_\mathbf{w}^{(1)}, \ldots, \tilde{h}_\mathbf{w}^{(m_L)})$.

Finally, we define the class

$$\mathcal{F}_{W,L} := \{\mathcal{F} = \mathcal{F}(m_0, \ldots, m_\ell) \mid \ell \leq L,\, W_L(\mathcal{F}) \leq W\}, \tag{34}$$

consisting of DNNs with at most $L$ hidden layers and a total number of weights not exceeding $W$.

**Lemma 8.** *Consider GCNNs where the G-correlation uses kernels from a one-dimensional vector space with a fixed basis given by the indicator function of the identity element $e$. For every $\tilde{h}_\mathbf{w} \in \mathcal{F}(m_0, \ldots, m_L)$, there exists a GCNN $h_\mathbf{w}$ with the same number of channels in each layer, i.e., $h_\mathbf{w} \in \mathcal{H}(1, m_0, \ldots, m_L, r)$, and parameters $\mathbf{w}$, such that for any input function $f : G^r \to \mathbb{R}^{m_0}$,*

$$\sum_{i=1}^{m_L}\sum_{j=1}^{r}\tilde{h}_\mathbf{w}^{(i)}(f(g_j)) = h_\mathbf{w}(f).$$

*Proof.* Write $\mathcal{H} := \mathcal{H}(1, m_0, \ldots, m_L, r)$. Consider a fixed input function $f : G^r \to \mathbb{R}^{m_0}$ and a weight vector $\mathbf{w} \in \mathbb{R}^W$. Recall that the number of parameters in a GCNN is given by

$$W_L := \sum_{j=1}^{L} m_j(k m_{j-1} + 1), \tag{35}$$

where $k$ is the dimension of the kernel space. In our case $k = 1$ and the number of parameters for a GCNN with architecture $\mathcal{H}$ is

$$W_L = \sum_{j=1}^{L} m_j(m_{j-1} + 1). \tag{36}$$

This coincides with the number of parameters in a DNN with architecture $\mathcal{F}(m_0, \ldots, m_L)$. Consequently, the same weight vector $\mathbf{w} \in \mathbb{R}^W$ defines both, a DNN function $\tilde{h}_{\mathbf{w}}$ and a GCNN function $h_{\mathbf{w}} \in \mathcal{H}$ when the input is fixed to $f$.

We now show that the outputs of the computational units in $\tilde{h}_{\mathbf{w}}$ and $h_{\mathbf{w}}$ are equal when applied to $f(g)$ and $g$, respectively. Specifically, we aim to prove that

$$\tilde{h}_{\ell,i}\big(f(g_j)\big) = h_{\ell,i}(g_j),$$

where $\tilde{h}_{\ell,i}$ denotes a DNN computational unit in layer $\ell$ of $\tilde{h}_{\mathbf{w}}$, with parameters fixed to $\mathbf{w}$, and $h_{\ell,i}$ represents a GCNN computational unit in layer $\ell$, with parameters fixed to $\mathbf{w}$ and input set to $f$. We prove this by induction on the layer $\ell$.

The statement holds trivially for the input layer, as $\tilde{h}_{0,i}(f(g_j)) = h_{0,i}(g_j)$ for any $g_j \in G^r$. Assuming it holds for all layers up to $\ell - 1$, we now prove it for layer $\ell$.

Let $\mathtt{K}$ denote the indicator of the identity element $e \in G^r$. By calculating the G-correlation between $\mathtt{K}$ and $f$, we obtain $\mathtt{K} * f = f$. Combining this with the definition of the GCNN unit (see (9) in the main article) and the induction hypothesis, we have

$$
\begin{aligned}
\tilde{h}_{\ell,i}(g_j) &:= \sigma\left( \sum_{t=1}^{m_{\ell-1}} \mathbf{w}_{t,i}^{(\ell-1)} \tilde{h}_{\ell-1,t}(g_j) - b_i^{(\ell)} \right) \\
&= \sigma\left( \sum_{t=1}^{m_{\ell-1}} \mathbf{w}_{t,i}^{(\ell-1)} h_{\ell-1,t}(g_j) - b_i^{(\ell)} \right) && \text{(induction assumption)} \\
&= \sigma\left( \sum_{t=1}^{m_{\ell-1}} \left( \mathbf{w}_{t,i}^{(\ell-1)} \mathtt{K} * h_{\ell-1,t} \right)(g_j) - b_i^{(\ell)} \right) && \text{(property of } \mathtt{K}) \\
&= \sigma\left( \sum_{t=1}^{m_{\ell-1}} \left( \mathcal{K}_{\mathbf{w}_{t,i}^{(\ell-1)}} * h_{\ell-1,t} \right)(g_j) - b_i^{(\ell)} \right) && \text{(definition of learned kernel)} \\
&= h_{\ell,i}(g_j) && \text{(definition of GCNN unit)}.
\end{aligned}
$$

This shows that the outputs of the computational units in $\tilde{h}_{\mathbf{w}}$ and $h_{\mathbf{w}}$ are equal when applied to $f(g)$ and $g$, respectively.

Finally, the outputs of $\tilde{h}_{\mathbf{w}} := (\tilde{h}_{\mathbf{w}}^{(1)}, \ldots, \tilde{h}_{\mathbf{w}}^{(m_L)})$ can be rewritten into the form

$$\sum_{i=1}^{m_L} \sum_{j=1}^{r} \tilde{h}_{\mathbf{w}}^{(i)}(f(g_j)) = \sum_{i=1}^{m_L} \sum_{j=1}^{r} \tilde{h}_{L,i}(g_j) = \sum_{i=1}^{m_L} \sum_{j=1}^{r} h_{L,i}(g_j) = h_{\mathbf{w}}(f),$$

concluding the proof of the lemma. $\qquad\square$

Next, we prove Lemma 2. The key ideas and steps of the proof have already been outlined in the main article, so here we will focus on the formal statements that still needs to be established.

Recall that the indicator neural network

$$\mathbf{1}_{(a,b,\epsilon)} \tag{37}$$

is a shallow ReLU network with four neurons in the hidden layer (see (25) in the main article). It approximates the indicator function on the interval $[a, b]$ in the sense that $\mathbf{1}_{(a,b,\epsilon)}(x) = 1$ if $x \in [a, b]$, and $\mathbf{1}_{(a,b,\epsilon)}(x) = 0$ if $x < a - \epsilon$ or $x > b + \epsilon$.

**Lemma 9.** *[Lemma 2] For $L > 3$ let $\mathcal{H}_{6W,L+1,r}$ be the class of GCNNs defined as in (33) and $\mathcal{F}_{W,L}$ be the class of DNNs defined as in (34). Then*

$$\mathrm{VC}(\mathcal{H}_{6W,L+1,r}) \geq \mathrm{VC}(\mathcal{F}_{W,L}).$$

*Proof.* Let $m$ be the VC dimension of the class of DNNs $\mathcal{F}_{W,L}$. There are $2^m$ possible binary classifications for a set of $m$ elements, subsequently denoted by $d = 2^m$.

By definition, there exists a natural number $m_0$ and a set of $m$ vectors

$$\mathcal{Y} := \{\mathbf{y}_1, \ldots, \mathbf{y}_m\} \subset \mathbb{R}^{m_0}, \tag{38}$$

that can be shattered by a subset of networks $\tilde{\mathcal{H}} \subset \mathcal{F}_{W,L}$. Since there are no more than $d$ distinct classifiers for $\mathcal{Y}$, the class $\tilde{\mathcal{H}}$ consists of at most $d$ DNN functions.

Next, we construct a DNN architecture using $m_0 + 1$ smaller DNN classes. One of these classes is $\tilde{\mathcal{H}}$, while the remaining $m_0$ classes consist of "indicator" networks, as described in (37). These indicator networks ensure that the combined DNN vanishes outside a certain $m_0$-dimensional hypercube. To define this hypercube, we use the set $\mathcal{Y}$ from above.

Specifically, we choose numbers $A > \max_{\mathbf{y} \in \mathcal{Y}} \|\mathbf{y}\|_\infty + 1$ and $B > A$, and define the $m_0$-dimensional hypercube
$$\Pi := \{\mathbf{y} = (y_1, \ldots, y_{m_0}) \mid A \leq y_i \leq B\}.$$

To construct a DNN that vanishes on $\Pi$, we define an approximate indicator function for $\Pi$, using a DNN with $m_0$-dimensional input $\mathbf{y} = (y_1, \ldots, y_{m_0})$:
$$I : \mathbb{R}^{m_0} \to \mathbb{R}, \quad I(\mathbf{y}) := \frac{1}{m_0} \sum_{i=1}^{m_0} \mathbf{1}_{(A,B,0.5)}(y_i),$$

where $\mathbf{1}_{(A,B,0.5)}(y_i)$ is an indicator network that approximates the indicator function for values within $(A, B)$.

The final DNN is formed by combining functions from $\tilde{\mathcal{H}}$ with the indicator function $I$. Since DNNs can be summed if they have the same depth, we adjust the depth of $I$ to match the depth of the functions from $\tilde{\mathcal{H}}$ while ensuring that $I$ remains constant on $\mathcal{Y}$ and $\Pi$. Specifically, we use the fact that for $I(\mathbf{y}) > 0$, $\sigma(I(\mathbf{y})) = I(\mathbf{y})$ for the ReLU activation function $\sigma(x) = \max\{x, 0\}$ (for any $\mathbf{y}$ from $\Pi$ or $\mathcal{Y}$). This means that by composing $I$ with the required number of ReLU functions, we can construct a DNN that satisfies the desired properties. This construction requires at most $L < W$ additional weights.

To complete the proof, we need to show that there are $m$ input functions $F_m := \{f_1, \ldots, f_m\} \subset \{f : G^r \to \mathbb{R}^{m_0}\}$ that can be shattered by GCNNs from $\mathcal{H}_{6W,L+1,r}$. As the set $F_m$, we choose functions defined by $f_i(e) = \mathbf{y}_i$ and $f_i(g) \in \Pi$ for $g \in G^r$ and $g \neq e$.

By the definition of shattering (see definition in the main article), to prove that $F_m$ is shattered, it is sufficient to show that for any binary classifier $\mathcal{C} : F_m \to \{0, 1\}$, there exists a corresponding function in $\text{sign}(\mathcal{H}_{6W,L+1,r})$ whose values coincide with those of $\mathcal{C}$ on $F_m$.

Choose $\tilde{h} \in \tilde{\mathcal{H}}$ such that for some $b \in \mathbb{R}$, $\text{sign}(\tilde{h}(\mathbf{y}_i) - b) = \mathcal{C}(f_i)$ for $i = 1, \ldots, m$.

Since $\Pi$ is compact, we define
$$T := \max_{\mathbf{y} \in \Pi} |\tilde{h}(\mathbf{y})|.$$

The final DNN $\tilde{h}_{\mathcal{C}}$ adjusts $\tilde{h}$ such that it vanishes on $\Pi$ but coincides with $\text{sign}(\tilde{h} - b)$ on $\mathcal{Y}$,
$$\tilde{h}_{\mathcal{C}} := \sigma(\tilde{h} - (T - b)I - b),$$

with $\sigma(x) = \max\{x, 0\}$

Thus, for any $f_i \in F_m$,
$$\text{sign}\left(\sum_{j=1}^{r} \tilde{h}_{\mathcal{C}}(f_i(g_j))\right) = \text{sign}(\tilde{h}(\mathbf{y}_i) - b) = \mathcal{C}(f_i).$$

By Lemma 8, we can define a GCNN $h_{\mathcal{C}}$ such that $h_{\mathcal{C}}(f) = \sum_{s=1}^{r} \tilde{h}_{\mathcal{C}}(f(g_s))$ for any $f \in F_m$. This implies that $\text{sign}(h_{\mathcal{C}}(f_i - b)) = \mathcal{C}(f_i)$ for any $f_i \in F_m$.

As the number of weights in $\tilde{h}_{\mathcal{C}}$ is $W + L + 4m_0 < 6W$, this shows that our GCNN is in the class $\mathcal{H}_{6W,L+1,r}$, completing the proof of the lemma. $\square$

**Corollary 4.** *[Corollary 2] In the setting of Lemma 2, if the number of layers $L > 3$ and $L \leq W^{0.99}$, then there exists a constant $c$ such that*
$$\text{VC}(\mathcal{H}_{W,L,r}) \geq c \cdot \text{VC}(\mathcal{F}_{W,L}).$$

*Proof.* From Equation (2) in [3] , we know that for the class of fully connected neural networks $\mathcal{F}_{W,L}$ with $L$ layers and at most $W$ overall parameters, there exist constants $c_0$ and $C_0$ such that

$$c_0 \cdot WL \log\left(\frac{W}{L}\right) \leq \text{VC}(\mathcal{F}_{W,L}) \leq C_0 \cdot WL \log W. \tag{39}$$

Moreover, by Lemma 9, we have

$$\text{VC}(\mathcal{H}_{6W',L'+1,r}) \geq \text{VC}(\mathcal{F}_{W',L'}).$$

By choosing $W = 6W'$ and $L = L' + 1$, this shows that

$$\text{VC}(\mathcal{H}_{W,L,r}) \geq \text{VC}(\mathcal{F}_{\lfloor \frac{1}{6}W \rfloor, L-1}).$$

To obtain the statement in the lemma, we combine this bound with the left inequality in (39), leading to

$$\text{VC}(\mathcal{H}_{W,L,r}) \geq \text{VC}(\mathcal{F}_{\lfloor \frac{1}{6}W \rfloor, L-1}) \geq c_0 \cdot \left(\frac{1}{6}W - 1\right)(L-1)\log\left(\frac{\frac{1}{6}W - 1}{L-1}\right).$$

For some constant $c_1 > 0$, the right-hand side of this inequality is bounded from below by

$$c_1 \cdot WL \log W.$$

By using the right inequality in (39), this can be further bounded,

$$c_1 \cdot WL \log W \geq c \cdot \text{VC}(\mathcal{F}_{W,L}),$$

showing the assertion. $\qquad\square$

Next, we provide the proof for the second part of Theorem 2, which states that for some universal constant $c > 0$, the VC dimension $\text{VC}(\mathcal{H}_{W,L,r})$ is bounded by $c \cdot W \log_2(r)$. As mentioned in the main article, the first step of the proof is Lemma 3.

**Lemma 10.** *[Lemma 3] Let $\mathcal{H}_{4,L,r}$ be the class of GCNNs defined in* (33)*. Then*

$$VC(\mathcal{H}_{4,L,r}) \geq \lfloor \log_2 r \rfloor.$$

*Moreover, for any two numbers $A < B$, there exists a finite subclass of GCNNs $\mathcal{H} \subset \mathcal{H}_{4,L,r}$ that shatters a set of $\lfloor \log_2 r \rfloor$ input functions*

$$F_m := \{f_i : G^r \to [A, B] \mid i = 1, \ldots, \lfloor \log_2 r \rfloor\},$$

*and outputs zero for any input function $f : G^r \to \mathbb{R} \setminus [A, B]$.*

*Proof.* To simplify the notation, let $m := \lfloor \log_2 r \rfloor$. It will be enough to show that a subclass of GCNNs $\mathcal{H} \subset \mathcal{H}_{4,L,r}$ shatters the set of $m$ input functions as this immediately implies that

$$VC(\mathcal{H}_{4,L,r}) \geq \lfloor \log_2 r \rfloor.$$

The proof involves selecting $d := 2^m$ distinct points in the interval $[A, B]$ and defining "indicator" neural networks of the form (37) that output 1 at exactly one of these points. By adjusting the parameters of these networks, we can control the intervals of our indicator networks and ensure that each network outputs 1 at the desired point.

Specifically, define $\delta := \frac{B-A}{2(d+2)}$ and select the $d$ points

$$\mathcal{Y} := \{y_i := A + i\delta \mid i = 1, \ldots, d\}.$$

The input functions $F_m$ are now chosen from $\{f : G^r \to \mathcal{Y} \cup \{B - \delta\}\}$.

There are $d = 2^m$ different binary classifiers for the set of $m$ elements. Each binary classifier is defined by the elements for which it outputs 1, and we can index these classifiers by the subsets of $\{1, 2, \ldots, m\}$, denoted by $S_1, \ldots, S_d$. In our construction, each $y_i \in \mathcal{Y}$ corresponds to the binary classifier determined by $S_i$. More formally, the set of $m$ input functions $F_m := \{f_1, \ldots, f_m\}$ is defined by

$$f_j(g_i) := \begin{cases} y_i, & \text{if } j \in S_i, \\ B - \delta, & \text{otherwise.} \end{cases}$$

Next, we define the finite subclass in $\mathcal{H}_{4,L,r}$ that shatters $F_m$ and outputs zero for any function $f : G^r \to \mathbb{R} \setminus [A, B]$.

By the definition of shattering (definition is in the main article), for any binary classifier $\mathcal{C} : F_m \to \{-1, 1\}$, we need to find a function in $\text{sign}(\mathcal{H}_{4,L,r})$ matching $\mathcal{C}$ on $F_m$.

For any classifier $\mathcal{C} : F_m \to \{-1, 1\}$ we can find a subset $S \subseteq \{1, \ldots, m\}$ such that $\mathcal{C}(f_j) = 1$ if $j \in S$ and $\mathcal{C}(f_j) = -1$ if $j \in S^c$. There exists an index $i^*$ such that $S = S_{i^*}$. Using Lemma 8, one can construct a GCNN $h_{i^*} \in \mathcal{H}_{4,L,r}$ that matches $\mathcal{C}$ on $F_m$. Indeed, for any $f_j \in F_m$,

$$h_{i^*}(f_j) := \sum_{s=1}^{r} \mathbf{1}_{(y_{i^*} - \frac{\delta}{2}, y_{i^*} + \frac{\delta}{2}, \frac{\delta}{2})}\big(f_j(g_s)\big) = \begin{cases} 1, & \text{if } j \in S_{i^*}, \\ 0, & \text{otherwise.} \end{cases} \tag{40}$$

Thus, $\text{sign}(h_{i^*}(f) - 0.5) = \mathcal{C}(f)$ for all $f \in F_m$. As an 'indicator' neural network $\mathbf{1}_{(y_{i^*} - \frac{\delta}{2}, y_{i^*} + \frac{\delta}{2}, \frac{\delta}{2})}$ has only 4 parameters and 2 layers, it is in $\mathcal{H}_{4,L,r}$.

Moreover, for any $i = 1, \ldots, d$ and any $x \in \mathbb{R} \setminus [A, B]$, $\mathbf{1}_{(y_i - \frac{\delta}{2}, y_i + \frac{\delta}{2}, \frac{\delta}{2})}(x) = 0$. Arguing as for (40), $h_{i^*}(f) = 0$ for any $f : G^r \to \mathbb{R} \setminus [A, B]$.

That means that the class $\mathcal{H} := \{h_1, \ldots, h_d\}$ shatters input functions $F_m$ and outputs 0 on the subset $\{f : G^r \to \mathbb{R} \setminus [A, B]\}$. This completes the proof. $\qquad\square$

**Corollary 5.** *[Corollary 3] The VC dimension of the class $\mathcal{H}_{4W,L,r}$, consisting of GCNNs with $4W$ weights, $L$ layers, and resolution $r$ satisfies the inequality*

$$\text{VC}\big(\mathcal{H}_{4W,L,r}\big) \geq W \lfloor \log_2 r \rfloor.$$

*Proof.* To simplify notation, let $m := \lfloor \log_2 r \rfloor$.

We prove this corollary by defining $W$ disjoint intervals $[A_1, B_1], \ldots, [A_W, B_W]$, where $A_i := (m + 3)i$ and $B_i := (m + 2)i$. For different $i \in \{1, \ldots, W\}$ the set of input functions $\mathcal{F}_i := \{f : G^r \to [A_i, B_i]\}$ is disjoint since the values of the intervals do not overlap.

By Lemma 10, for each $i = 1, \ldots, W$, we can find a class of GCNNs $\mathcal{H}_i \subset \mathcal{H}_{4,L,r}$ that shatters a set of $m$ input functions $F_{m,i} \subset \mathcal{F}_i$ and outputs 0 on any other set $F_{m,j}$, where $j \neq i$.

Next we show that the class of GCNNs $\mathcal{H} := \mathcal{H}_1 \oplus \mathcal{H}_2 \oplus \cdots \oplus \mathcal{H}_W \subset \mathcal{H}_{4W,L,r}$ shatters the set $F_{Wm} := \bigsqcup_{i=1}^{W} F_{m,i}$. This will prove the corollary.

By the definition of shattering, we need to find for any binary classifier $\mathcal{C} : F_{Wm} \to \{0, 1\}$, a function in $\text{sign}(\mathcal{H})$ that matches $\mathcal{C}$ on $F_{Wm}$.

For $i = 1, \ldots, W$, let $\mathcal{C}_i := \mathcal{C}|_{F_{m,i}}$ be the restriction of $\mathcal{C}$ to $F_{m,i}$. As $\mathcal{H}_i$ shatters $F_{m,i}$, we can choose a GCNN $h_{\mathcal{C}_i} \in \mathcal{H}_i$ such that its values match those of $\mathcal{C}_i$ on $F_{m,i}$.

Next, we show that the values of the GCNN $h_{\mathcal{C}} := \sum_{i=1}^{W} h_{\mathcal{C}_i}$ match $\mathcal{C}$ on $F_{Wm}$. Let $f$ be any input function from $F_{Wm}$, say $f \in F_q$. For any $i \neq q$, it holds that $h_{\mathcal{C}_i}(f) = 0$ since $h_{\mathcal{C}_i} \in \mathcal{H}_i$. Thus,

$$\sum_{i=1}^{W} h_{\mathcal{C}_i}(f) = h_{\mathcal{C}_q}(f).$$

Since $h_{\mathcal{C}_q}(f) = \mathcal{C}(f)$ by the choice of $h_{\mathcal{C}_q}$, it follows that $h_{\mathcal{C}}(f) = \mathcal{C}(f)$ for any $f \in F_{Wm}$.

This shows that the class $\mathcal{H}$ of GCNNs shatters $F_{Wm}$, proving the corollary. $\qquad\square$

## C  On the constants in Equation (1)

While our main focus is on the asymptotic behavior of the VC dimension, it is instructive to express the constants $c$ and $C$ for GCNNs in terms of the corresponding constants for DNNs. For DNNs, the Eq(2) from [3] states that there exist universal constants $c_{\mathcal{F}}$ and $C_{\mathcal{F}}$ such that

$$c_{\mathcal{F}} \cdot WL \log\left(\frac{W}{L}\right) \leq \text{VC}(\mathcal{F}_{W,L}) \leq \max_{\mathcal{F} \in \mathcal{F}_{W,L}} UB(\mathcal{F}) \leq C_{\mathcal{F}} \cdot WL \log W, \tag{41}$$

where $UB(\mathcal{F})$ denotes an upper bound on the VC dimension for the class $\mathcal{F}$. As we show now, under the commonly satisfied assumptions that $W > L^2$ and $W > 125$, we obtain

$$c_{\mathcal{F}} = 0.04, \quad C_{\mathcal{F}} = 5, \quad c = \frac{1}{7500}, \quad C = 500.$$

To derive an explicit lower bound for the constant $c_{\mathcal{F}}$ for the class $\mathcal{F}$ of DNNs with at most $W$ weights and $L$ layers, we follow Theorem 9 [3]. This theorem constructs networks with VC dimension $\geq mn$, where the architecture has $3 + 5k$ layers,

$$2 + n + 4m + k\big((11 + r)2^r + 2r + 2\big) \text{ network parameters,}$$

$m + n$ input nodes, and

$$m + 2 + k\left(5 \times 2^r + r + 1\right)$$

computational nodes. For the parameter choices

$$r = \frac{\log_2\left(\frac{W}{L}\right)}{2}, \quad m = \frac{rL}{8}, \quad n = W - 5m \cdot 2^r,$$

the VC dimension satisfies

$$VC(\mathcal{F}) \geq mn = \frac{rL}{8}(W - 5m \cdot 2^r) = \frac{WL\log_2\left(\frac{W}{L}\right)}{16}\left(1 - \frac{5\log_2\left(\frac{W}{L}\right)}{16}\sqrt{\frac{L}{W}}\right)$$

To refine the estimate further, we use the assumption that $W > L^2$ and $W > 125$, such that

$$VC(\mathcal{F}) \geq \frac{WL\log_2\left(\frac{W}{L}\right)}{16}\left(1 - \frac{5\log_2(\sqrt{W})}{16}\frac{1}{W^{1/4}}\right)$$

$$\geq \frac{WL\log_2\left(\frac{W}{L}\right)}{16}0.674 > 0.04WL\log_2\left(\frac{W}{L}\right).$$

That means that $c_{\mathcal{F}}$ can be chosen as $0.04$.

To derive an explicit value for $C_{\mathcal{F}}$, we follow the proof of Theorem 6 from [3]. In that proof, it is stated that:

$$\text{VCdim}(\mathcal{F}) \leq L + \left(\sum_i W_i\right)\log_2\left(4epR\log_2(2epR)\right)$$

where

$$R \leq W + W(L-1)d^{L-1}.$$

In the case of the ReLU activation function, we have $p = d = 1$ and so,

$$\text{VCdim}(\mathcal{F}) \leq L + \left(\sum_i W_i\right)\log_2\left(4epR\log_2(2epR)\right)$$

$$\leq L + 2WL\log_2\left(4epR\right)$$
$$\leq L + 2WL\log_2\left(4eWL\right)$$
$$\leq L + 2WL\log_2\left(4eW^{3/2}\right)$$
$$\leq L + 3WL\log_2\left(4eW\right)$$

Under the assumption that $W > L^2$ and $W > 125$, we have

$$L + WL\log_2(4e) < L + 1.5WL\log_2(125) \leq L + 1.5WL\log_2(W) < 2WL\log_2(W)$$

leading us to

$$\text{VCdim}(\mathcal{F}) \leq 5WL\log_2(W),$$

and $C_{\mathcal{F}} = 5$. Therefore $C_{\mathcal{F}}/c_{\mathcal{F}} = 5/0.04 = 125$.

To derive $C$ and $c$, recall that

$$c_{\mathcal{F}} \cdot WL\log(W/L) \leq VC(F_{W,L}) \leq UB(F_{W,L}) \leq C_{\mathcal{F}} \cdot WL\log W. \tag{2}$$

Imposing again the assumptions that $W > L^2$ and $W > 125$, we have

$$UB(F_{W,L}) \leq \frac{C_{\mathcal{F}}}{c_{\mathcal{F}}} \cdot 2 \cdot c_{\mathcal{F}} WL \log(W/L) = 250 \text{VC}(F_{W,L}).$$

Using this and Eq. (19), we obtain

$$\text{VC}(\mathcal{H}_{W,L,r}) \leq 2\,UB(F_{W,L}) + 8WL \log r \leq 500\big(\text{VC}(F_{W,L}) + WL \log r\big).$$

For the lower bound, using Lemma 2, we estimate

$$\text{VC}(\mathcal{F}_{\lceil \frac{W}{5} \rceil, L-1}) \geq \frac{1}{5} c_{\mathcal{F}} W(L-1) \log \left( \frac{W}{5(L-1)} \right) \geq \frac{1}{60} c_{\mathcal{F}} WL \log W$$

$$\geq \frac{1}{60} \cdot \frac{c_{\mathcal{F}}}{C_{\mathcal{F}}} \cdot \text{VC}(\mathcal{F}_{W,L}) \geq \frac{\text{VC}(\mathcal{F}_{W,L})}{7500}.$$

Under stronger assumptions (e.g., $W \geq L^3$), tighter bounds can be derived.

