# OpenReview forum: "On the VC dimension of deep group convolutional neural networks"
_NeurIPS.cc/2025/Conference — NeurIPS 2025 poster_

### Official Review · Reviewer_29km · 2025-06-12

**Clarity:** 4
**Significance:** 3
**Originality:** 3
**Rating:** 5
**Confidence:** 4

**Summary:**

This paper studies the generalization capacity of group convolutional neural networks. It identifies an insightful relationship between the VC dimension of a set of fully connected neural networks and group convolutional neural networks (Equation 1).
Intriguingly, the GCNNs have a larger VC dimension than their fully connected counterparts, even if the same number of parameters are assumed. This sheds light on the effectiveness of CNNs and corresponds well to empirical observations.

**Questions:**

I do not have any questions. I feel like I have fully understood the manuscript. The authors could only increase my score by picking a significantly harder problem and solve that.

**Ethical Concerns:**

["NO or VERY MINOR ethics concerns only"]

**Final Justification:**

It seems that the authors agree with my initial report.

**Limitations:**

yes

**Quality:**

4

**Strengths And Weaknesses:**

I think the writing is clear, and the mathematical arguments are sound. The manuscript is very focused on the main point---the VC dimension bounds described in Equation 1. The title, abstract, and introduction all appropriately describe the results, without oversimplification or exaggeration.

The only weakness is that, while the results are interesting, I would not describe them as ground-breaking. Still, this is a very good paper.

---

### Official Review · Reviewer_4cFh · 2025-06-20

**Clarity:** 3
**Significance:** 3
**Originality:** 3
**Rating:** 4
**Confidence:** 3

**Summary:**

This paper provides upper and lower bounds for the VC dimension of group CNN (GNNs) showing that for finite groups that the VC dimension of a GCNN is roughly equivalent to that of an unstructured DNN with the same number of parameters / layers. It builds on existing work by showing considering networks of arbitrary depth.

Despite my concern about whether or not the VC dimension is the right tool to analyze equivariant ML, I still think this paper makes a major advance in our understanding. (Indeed this analysis is what motivates me to suggest we need a new notion of VC dimension in the comments below.) Overall, I think this is a good paper.

Note: My score is conditioned on the authors making all of the requested edits and I reserve the right to change it if they do not.

**Questions:**

Do we require G^r to be a subset of G?

Why isn't the definition of \mathcal{K}\ast f normalized in equation (5)

Where does the formula (16) come from?

In the proof of Lemma 1, where does the identity VC(F_r) \approx W(F_r)L log(W(F_r)) come from?

In the proof of Lemma 5, where does the "at most m_{ell+1}mr" polynomials come from?

**Ethical Concerns:**

["NO or VERY MINOR ethics concerns only"]

**Final Justification:**

I stand by my initial review and think this is a good paper

**Limitations:**

Yes

**Quality:**

3

**Strengths And Weaknesses:**

Strengths

VC dimension has traditionally played an important role in understanding model generalization. This work extends this to GCNNs thus increasing the theoretical foundations of geometric deep learning

The paper is generally well written and explained

Proofs are well organized and appear to be correct. Morever, the analysis is highly nontrivial. (I do admit that as someone who only has a cursory understanding of the VC dimension that I was not able to understand every step in its entirety.)

Weaknesses

It would be helpful to have a survey of a) why we care about the VC dimension, e.g., recalling the generalization bounds of ML algorithms that depend on it and b) what results are known for various DNN models, possibly in the appendix. This would help keep the paper self-contained for the reader.

I think the claim that K\ast g = r \cdot K \star g is an approximation not an equality. If not, please justify. (Maybe it is true in expectation if the elements of G^r are sampled randomly from the Haar measure?)

Comment:

I am a bit unsure if the VC dimension is the "right" way to understand the advantages of equivariant ML.  It essentially analyzes the ability to learn arbitrary functions on SOME data set with a given cardinality. In some sense, it is not surprising that the VC dimension would then be the same for equivariant / non-equivariant models. To me, it seems that the advantages of equivariant ML is that it restricts the model class to not be able to express functions with the wrong symmetries. Therefore, it seems that equivariant / non-equivariant methods should be able to express arbitrary functions on data sets of the same size as long as the data sets didn't include two points from the same orbit of the group action.

(continued) It might be better to have some notion of an "equivariant VC dimension" which analyzed the ability of a GCNN to shatter data in terms of the number of orbits present. Here, the ability of a DNN to express arbitrary functions would be only in terms of the number of distinct points, but the GCNN would be forced to be constant on orbits. In this manner the GCNN and DNN would have the same "standard" VC dimension but the GCNN would have a lower equivariant VC Dimension if the data had many "repeat data points" from the same orbit

Minor:

In the definition of a group, there is a "for all" missing in the definition of the inverse.

Line 207: proof sketch is in Section 5, not section 2

More detail should be given in the final inequality of Lemma 1. Also, the O should be a Theta

The m in the p_m(x) in Lemma 4 is missing a tilde

Title of Appendix A/B, theorem should be capitalized

---

> ### Author Rebuttal · Authors · 2025-07-30
>
> We thank the reviewer for the thoughtful and constructive feedback. We believe that these suggestions have improved the clarity and rigor of our work, and we look forward to incorporating them into the final version.
>
> - **Consider including a brief survey of: (a) why VC dimension matters, e.g., in generalization bounds; and (b) existing VC results for DNNs.**
>
>     Thank you for this suggestion. We agree that this addition would make the paper more accessible, especially for readers unfamiliar with the role of VC dimension in learning theory. We will include a brief overview and relevant references in the final version.
>
>     The main motivation for studying VC dimension is its role in generalization guarantees, as explained to Reviewer qLBm. Moreover, there is a close relation between the VC dimension bounds and the expressive power of neural networks. In particular, upper bounds on the VC dimension directly imply lower bounds on feasible approximation rates. Most of the known VC dimension bounds for DNNs are summarized in Anthony and Bartlett (1999). In Bartlett et al. (2019) tightness of VC dimension bounds for piecewise linear activation functions is shown. We will include a summary of known results in the revised version.
>
> - **It might be better to define an "equivariant VC dimension" based on the number of orbits in the data.**
>
>     Thank you for this insightful comment. It would indeed be valuable to define an "equivariant VC dimension" that accounts for the model's constraint to produce outputs that are constant on orbits. This could reflect more realistic limitations of GCNNs when data points are repeated across orbits.
>
>     Our paper focuses on the standard VC dimension. In this sense, GCNNs and DNNs have the same capacity when data lies on distinct orbits. However, under more structured data assumptions (e.g., repeated or orbit-related data), an equivariant VC dimension may better capture model expressivity. This is a promising direction for future work.
>
> - **Clarify the claim that $K \ast g = r \cdot K \star g$. Is this an approximation?**
>
>     This is indeed an equality under our definition of $\star$. Please refer to Equations (4) and (5) in the paper. We agree that this notation may be confusing, but using $\star$ helps to streamline technical proofs.
>
> - **Do we require $G^r \subseteq G$?**
>
>     Yes, we assume that $G^r \subseteq G$, as our convolutional filters are defined over $G$.
>
> - **Why isn't the definition of $\mathcal{K} \ast f$ normalized in Equation (5)?**
>
>     We dont have the normalization factor in Equation (5) to simplify the technical proofs. This simplification helps in the proofs of Lemmas 8--10 (Lemmas 2--3 in the main text).
>
>
> - **Where does Equation (16) come from?**
>
>     Equation (16) is derived by counting the total number of weights in the network. For each layer $j$, every computational unit computes an affine function over the outputs of the previous layer using $k$ kernel basis functions. This means that each unit in layer $j$ has $k m\_{j-1}$ weights (one for each combination of a kernel and a unit in layer $j - 1$), plus one bias term. Therefore, each unit contributes $k m\_{j-1} + 1$ weights, and with $m\_j$ units in layer $j$, the total number of weights between layers $j - 1$ and $j$ is $m_j (k m\_{j-1} + 1)$. Summing over all layers yields the total number of weights in the network, as derived in Equation (16).
>
> - **In the proof of Lemma 1, where does the identity $VC(F_r) \approx W(F_r) L \log W(F_r)$ come from?**
>
>     This follows from Bartlett et al. (2019), specifically Equation (2) and its corollary, under the assumption that $W^{0.99} > L$.
>
> - **In the proof of Lemma 5, where does the ``at most $m_{\ell+1} m r$'' polynomials come from?**
>
>     We are counting polynomials in the parameters of the neural network. To do so, we fix a computational unit in layer $\ell + 1$, a group element $g \in G^r$, and an input function $f \in F_m$. For each such triple, we obtain at most one distinct polynomial. Therefore, the total number of such polynomials is at most the product of the number of units in layer $\ell + 1$, the number of input functions in $F_m$, and the number of group elements in $G^r$, resulting in at most $m_{\ell+1} m r$ polynomials.

---

> > ### Comment · Reviewer_4cFh · 2025-08-01
> >
> > I confirm the response. I am satisfied with the response.

---

### Official Review · Reviewer_qLBm · 2025-06-23

**Clarity:** 3
**Significance:** 3
**Originality:** 3
**Rating:** 4
**Confidence:** 3

**Summary:**

This paper gives mathematical upper and lower bounds on the Vapnik-Chervonenkis (VC) dimension of deep Group Convolutional Neural Networks (GCNNs) with ReLU activations. The authors show that the VC dimension grows with the number of layers L, the total number of learnable weights W, and logarithmically with the size |G| of the discretized group that acts on the inputs. The bounds are almost tight and extend earlier work, which covered only two-layer GCNNs or continuous groups, to deeper architectures and practical finite groups. They also compare their results with known VC bounds for ordinary deep feed-forward networks and discuss sample-complexity consequences.

**Questions:**

1. How would your bounds will change if you choose other activations, for example, leaky-ReLU or smooth activations?
2. Have you estimated the inexplicit constants in the bounds to give practitioners guidance on required sample sizes?
3. Do the results extend to equivariant GCNNs that end with global max pooling instead of average pooling?
4. Could the logarithmic term in log |G| be improved if the group has additional structure (e.g., Abelian)?

**Ethical Concerns:**

["NO or VERY MINOR ethics concerns only"]

**Final Justification:**

Rebuttal convincingly covers leaky-ReLU and global max pooling; the lower-bound tweaks are sound. Still open: smooth activations (esp. upper bounds), constants not fully integrated, and no empirical checks. The novelty and near-tight theory for deep GCNNs outweigh these practicality gaps—hence borderline accept.

**Limitations:**

Yes

**Quality:**

3

**Strengths And Weaknesses:**

## Strengths
The main value of this paper is the theoretical aspects.  It gives the first almost-tight upper and lower bounds on the VC dimension of deep  GCNN built on a finite group (or discretized elements of continuous group).  The authors prove that the VC dimension rises linearly with the number of layers and weights, but only logarithmically with the size of the group, using the method of the polynomial partitioning.  This extends older results that covered only shallow networks. By writing their bounds as *VC dimension of a standard deep network + $W \log |G|$", they show clearly that adding  symmetry costs very little in a sample complexity.

## Weakness
The scope of this paper is relatively narrow:  all proofs assume that the activation is ReLU, and other activations remain open questions.  The work is purely theoretical, namely, there are no experiments or simulations to check how large the hidden constants are, so readers  cannot tell how many samples they really needs in practice.  While the bounds provides a hint that increasing the group resolution does not affect the size of VC dimension so much, the paper offers little practical advice on how to choose the resolution of the group and the number of weights.

---

> ### Author Rebuttal · Authors · 2025-07-31
>
> Thank you for your feedback. It is very helpful to further improve this submission.
>
> **How would your bounds change if you used other activations, such as leaky-ReLU or smooth activations?**
>
> The bounds remain the same for leaky-ReLU, as is the case for standard deep neural networks. The upper bound holds because leaky-ReLU is also a piecewise linear function with two linear segments, similar to ReLU. Thus, all arguments used in the proof of the upper bound apply directly.
>
> To justify that the lower bound still holds, we outline the necessary modifications to Lemma 2 and 3. We replace the indicator function by
> $$ \tilde{I}\_{A,B}(x) := \frac{1}{\epsilon} \Big[(x-(A-\epsilon))\_{+,\alpha} - (x-A)\_{+,\alpha} -
> \big((x-(B-\epsilon))\_{+,\alpha} - (x-B)\_{+,\alpha}\big)\Big], $$
>
> where $(x)_{+,\alpha} := \max(x,\alpha x)$, $0<\alpha<1$ is the Leaky ReLU parameter, $\epsilon > 0$, and $A < B-\epsilon$. This function is zero outside the interval $[A-\epsilon, B]$, and equal to 1 inside $[A, B-\epsilon]$.
>
> For Lemma 2, we use a fully connected network with leaky-ReLU activations. In our construction, we used a DNN that outputs zero for inputs outside a predefined hypercube and outputs a value $>1$ for inputs inside a smaller subcube. This property still holds for leaky ReLU activation.
>
> Specifically, we redefine the final function in Lemma 2 as:
> $$\tilde{h}\_{\mathcal{C}} \coloneqq \mathrm{LeakyReLU}\big( \tilde{h} - (T - b)\tilde{I}\_{A,B} - b \big) $$
>
> and select $f \in F_m$ such that $f(g) \in \Pi := {\mathbf{y} \in \mathbb{R}^{m_0} : A \leq y_i \leq B -\epsilon }.$
>
> In Lemma 3, we used the following construction: select $d = 2^m$ distinct points in $[A, B]$, and design networks that output 1 at exactly one point each. The use of $\tilde{I}_{A,B}$ remains valid if we choose the points in $[A, B-\epsilon]$ instead of $[A, B]$.
>
> For general smooth activation functions, the current proof strategy does not extend directly. While a lower bounds appears feasible due to the ability of sigmoidal networks to approximate the indicator function
> \begin{align*}
>     \sigma(k\cdot (x-A))-\sigma(k(x-B)) \approx \mathbf{1}_{[A,B]}(x), \quad \text{as} \ k \to \infty,
> \end{align*}
> one must carefully account for the fact that the network only approximates the binary outputs 0 and 1.
> In contrast, the upper bound does not follow readily. The region-counting argument, which applies to networks with piecewise polynomial activations like ReLU, breaks down for sigmoidal networks due to the absence of polyhedral decision boundaries. While alternative techniques have been used for sigmoid-activated networks, such as bounding the number of sign configurations of compositions of sigmoids and affine functions, these strategies do not directly extend to GCNNs and would require significant adaptation. Moreover, as noted by Bartlett and Anthony  (1999, Chapter 8), existing VC dimension bounds for deep neural networks with sigmoidal activations exhibit a substantial gap between known upper and lower bounds, indicating that even for vanilla DNNs the current results are not tight.
>
> **Have you estimated the constants in the bounds to help practitioners estimate required sample sizes?**
>
> We provided estimated constants in our response to Reviewer 6nZ2.
>
> In practice, the VC dimension relates to generalization through the fundamental theorem of learning theory, stating that
> $$m = O\left( \frac{\mathrm{VCdim}(\mathcal{H}) + \log(1/\delta)}{\varepsilon^2} \right),
> $$
> where $m$ is the required number of training samples to guarantee that the empirical loss approximates the true loss within $\varepsilon$ with probability at least $1 - \delta$. Thus, bounds on the VC dimension directly reveal the influence of the architecture on the learning complexity.
>
> **Do your results extend to equivariant GCNNs that end with global max pooling instead of average pooling?**
>
> Yes, the results still hold for global max pooling.
>
> For the lower bound, note that in Lemmas 2 and 3, the constructed examples have nonzero values only at a single group element. In such cases, global max and average pooling produce identical results, so the lower bound is preserved.
>
> Thank you for pointing this out. We will include in the revised version.
>
> **Could the logarithmic term in $\log |G|$ be improved if the group has additional structure (e.g., is Abelian)?**
>
> Our results hold for any finite group $G$, regardless of its algebraic structure. We make no assumptions beyond finiteness.

---

> > ### Comment · Reviewer_qLBm · 2025-08-09
> >
> > Thank you for addressing my concerns.

---

### Official Review · Reviewer_6nZ2 · 2025-07-03

**Clarity:** 3
**Significance:** 2
**Originality:** 2
**Rating:** 4
**Confidence:** 4

**Summary:**

The paper gives generalization bounds for certain Group Convolutional Neural Networks (GCNN), in particular cases in which the group is finite or which use a discretization by finitely many points of a (compact Lie) group. They have upper and lower bounds of VC dimension of GCNNs through the VC dimension of standard NNs with the same number of layers and weights per layer. The bounds involve a term in log(r) where r is the cardinality of the group. A special case is that of convolutional neural networks.

**Questions:**

I have just somehow minor questions.

1) What are the constants from equation (1)?

2) A curiosity: Does the approach give similar bounds on fat-shattering dimension?

3) Line 73: "groups such as rotations or translations are often discretized": then if the approximation of a Lie group is by a number of elements r that tends to infinity, will the bound (1) give infinite VC dimension as r tends to infinity? Or what is the version of the result for continuous compact Lie groups?

4) line 284 : "lemma 17" does not appear anywhere, what's the correct number?

**Ethical Concerns:**

["NO or VERY MINOR ethics concerns only"]

**Final Justification:**

After discussion with the authors, I consider that my assessment did not change considerably.

**Limitations:**

I don't think "societal impact" applies to this kind of result.

**Paper Formatting Concerns:**

No concern.

**Quality:**

3

**Strengths And Weaknesses:**

Strength: This is a new result on GCNN generalization bounds.

Strength: The result has upper and lower bounds, with equal growth behavior.

Weakness: The results only concern finite groups, and only describes VC and pseudodimension.

Overall I think that the strengths override the weaknesses. The result is interesting but I wouldn't go above "borderline accept" due to the weaknesses.

---

> ### Author Rebuttal · Authors · 2025-07-31
>
> We are very grateful for the time and effort spent on our submission. Please find a detailed point-by-point response below.
>
> **1. What are the constants from Equation (1)?**
>
> Our main focus is on asymptotic bounds. To obtain explicit expressions for the constants $c, C,$ it is necessary to carefully follow the proof steps. This also requires making the constants in Bartlett et al. (2019), _Nearly-tight VC-dimension and pseudodimension bounds for piecewise linear neural networks_ explicit. Let $c_{\mathcal{F}}$ and $C_{\mathcal{F}}$ denote the constants appearing in their VC dimension bounds for standard fully connected networks.
>
> Under the commonly satisfied assumptions that $W > L^2$ and $W > 125$, we obtain
> $$c = \frac{1}{120} \cdot \frac{c_{\mathcal{F}}}{C_{\mathcal{F}}}, \quad C = 4 \cdot \frac{C_{\mathcal{F}}}{c_{\mathcal{F}}}.$$
> Following the bounds in Bartlett et al. (2019), we can derive $c_{\mathcal{F}} = 0.04$ and $C_{\mathcal{F}} = 5.$ Substituting these values yields:
>
> $$c = 0.00006, \quad C = 500.$$
>
> A detailed derivation of these constants and their dependence on $c_\mathcal{F}$ and $C_\mathcal{F}$ will be included in the appendix of the final version.
>
> **2. A curiosity: Does the approach give similar bounds on the fat-shattering dimension?**
>
> Yes — for ReLU networks, the fat-shattering dimension and the pseudodimension coincide up to constant factors. This is because the ReLU activation is positively homogeneous: scaling the output of the last layer by a positive constant scales the whole network output.  Since $\mathcal{H}_{W,L,r}$ is closed under multiplication by positive scalars, the fat-shattering dimension coincides with the pseudodimension (see Theorem 11.14 in Anthony and Bartlett (1999)). Hence, our upper and lower bounds remain the same.
>
> **3. Line 73: What happens as $r \to \infty$ (or for continuous Lie groups)?**
>
> Yes, your interpretation is correct. If the group approximation becomes arbitrarily infinite (i.e., $r \to \infty$), and no regularity assumptions are made on the input data (e.g., continuity or smoothness with respect to the group action), then the VC dimension diverges to infinity. This also holds for continuous compact Lie groups. A discussion of this phenomenon appears also in Petersen et al.  (2024), _VC dimensions of group convolutional neural networks_ .
>
> **4. Line 284: ``Lemma 17'' does not appear anywhere — what's the correct number?**
>
> Thank you for catching this typo. The correct reference is **Lemma 8.** We will fix this in the revised version.

---

> > ### Comment · Reviewer_6nZ2 · 2025-08-01
> > **Thanks for the replies**
> >
> > Thank you for engaging with my questions and for the extra references.

---

### Decision · Program_Chairs · 2025-09-17

**Decision:**

Accept (poster)

**Comment:**

The paper establishes nearly tight upper and lower bounds for the VC dimension of deep group convolutional neural networks (GCNNs) with ReLU activations, showing that their growth matches that of standard deep networks in terms of layers and weights, with only a logarithmic dependence on the group size. This result extends prior work on shallow or continuous-group networks, offering the first general treatment for deep architectures on finite or discretized groups. Strengths include the novelty of the contribution, the clarity and rigor of the proofs, and the fact that the results capture both upper and lower bounds with matching growth, thereby solidifying the theoretical foundations of geometric deep learning. Reviewers appreciated the writing, the nontrivial analysis, and the insight that group symmetries increase sample complexity only marginally. Weaknesses noted include the narrow scope (results apply only to finite or discretized groups, with ReLU activations, and only in terms of VC and pseudo-dimensions), the lack of experimental validation or discussion of hidden constants (leaving practical implications unclear), and the limited broader impact since VC dimension may not fully capture the advantages of equivariant learning. Some reviewers suggested extensions (such as fat-shattering dimension, other activations, different pooling choices, and structured groups (e.g., Abelian) ) and one reviewer raised the need for an “equivariant VC dimension” that accounts for orbit structure (a related notion is discussed in this paper by Farrell et al. arxiv: 2110.07472 -- although not directly applicable to the problem here). Minor issues were noted in definitions, references, and clarity of certain proofs. Overall, the paper is well written, mathematically sound, and makes a meaningful though not ground-breaking advance; reviewers converge around a borderline-accept to accept recommendation.

Overall, I concur with the reviewers that the paper makes a good step in the theoretical picture around GCNNs, and would lean towards acceptance.